# A Review of Chitosan and Chitosan Nanofiber: Preparation, Characterization, and Its Potential Applications

**DOI:** 10.3390/polym15132820

**Published:** 2023-06-26

**Authors:** Marwan A. Ibrahim, Mona H. Alhalafi, El-Amir M. Emam, Hassan Ibrahim, Rehab M. Mosaad

**Affiliations:** 1Department of Biology, College of Science, Majmaah University, Al-Majmaah 11952, Saudi Arabia; 2Faculty of Women for Arts, Science and Education, Ain Shams University, Cairo 11566, Egypt; 3Department of Chemistry, College of Science, Majmaah University, Al-Majmaah 11952, Saudi Arabia; 4Faculty of Applied Arts, Textile Printing, Dyeing and Finishing Department, Helwan University, Cairo 11795, Egypt; 5Pretreatment and Finishing of Cellulosic Fibers Department, Textile Research and Technology Institute, National Research Centre, Cairo 12622, Egypt

**Keywords:** chitosan, nanofiber, preparation, characterization, applications, wound dressing

## Abstract

Chitosan is produced by deacetylating the abundant natural chitin polymer. It has been employed in a variety of applications due to its unique solubility as well as its chemical and biological properties. In addition to being biodegradable and biocompatible, it also possesses a lot of reactive amino side groups that allow for chemical modification and the creation of a wide range of useful derivatives. The physical and chemical characteristics of chitosan, as well as how it is used in the food, environmental, and medical industries, have all been covered in a number of academic publications. Chitosan offers a wide range of possibilities in environmentally friendly textile processes because of its superior absorption and biological characteristics. Chitosan has the ability to give textile fibers and fabrics antibacterial, antiviral, anti-odor, and other biological functions. One of the most well-known and frequently used methods to create nanofibers is electrospinning. This technique is adaptable and effective for creating continuous nanofibers. In the field of biomaterials, new materials include nanofibers made of chitosan. Numerous medications, including antibiotics, chemotherapeutic agents, proteins, and analgesics for inflammatory pain, have been successfully loaded onto electro-spun nanofibers, according to recent investigations. Chitosan nanofibers have several exceptional qualities that make them ideal for use in important pharmaceutical applications, such as tissue engineering, drug delivery systems, wound dressing, and enzyme immobilization. The preparation of chitosan nanofibers, followed by a discussion of the biocompatibility and degradation of chitosan nanofibers, followed by a description of how to load the drug into the nanofibers, are the first issues highlighted by this review of chitosan nanofibers in drug delivery applications. The main uses of chitosan nanofibers in drug delivery systems will be discussed last.

## 1. Introduction

Henry Braconot (1780–1855) isolated chitin from mushrooms in 1811. Chitin was the second carbohydrate discovered in 1859 [1]. It was hydrolyzed in an alkaline medium in 1894 to produce a new carbohydrate soluble in dilute acids called chitosan. Chitosan and its oligomers drew a lot of attention because of their unique properties [2,3,4]. Chitin represents the second natural polysaccharide in nature, after cellulose. Its chemical name is poly *N*-acetamido-2-decoxy-β-d-glucose [5]. Figure 1 shows the chemical structure and some sources of chitin and chitosan [6,7,8,9].

## 2. Structure of Chitin and Chitosan

Chitosan, a white polymer found in crabs, is a nitrogenous polysaccharide that is rigid inelastic and found both inside and outside the exoskeleton of invertebrates. The primary distinction is the solubility of chitosan in diluted acid (at PH = 7). The ocean, lakes, and seas are the sources of chitosan products [10]. Chitosan has potential applications in various fields, such as biomedical applications and the fabrication of protective clothes [10].

Chitin and chitosan, as natural renewable biopolymers, have unique properties, as they are biocompatible, biodegradable, and non-toxic [11]. Because of the NH_2_ groups that open the structure of chitosan, it is easier to modify than cellulose. As a result, it can be converted into a variety of chitosan derivatives [12]. The main purpose of this modification is to improve its solubility and chemical reactivity [13,14].

Acetylation degree is the presence of acetyl glucose amine to glucose amine in the chitin structure. Which determines the solubility of chitin and chitosan? Chitosan is more accepted than chitin due to its high degree of solubility in dilute acetic acid [15,16].

The presence of acetyl groups in chitin and chitosan makes their solubility problematic, although M.W.T. represents an important later component for chitin and chitosan properties. There are several methods used to estimate the molecular weight (M.W.T.) of chitin and chitosan, such as light scattering and viscosity [15,16].

Chitin is hydrophobic material insoluble in water and most organic solvents. Its solvent is a mixture of 1,2-diehloroethane and trichloroacetic acid (35:65), fresh saturated solution of lithium thiocyanate, chloroalcohols in conjugation with aqueous solutions of mineral acids and Dimethylacetamide containing 5% lithium chloride [10,17].

Chitin and chitosan are biocompatible, biodegradable, and non-toxic. Because they are amino polysaccharides, they exhibit interesting properties in biology, pharmacology, and physiology and have numerous applications such as wound healing, wound dressing, hemostatic agency, and antimicrobial activity, requiring them to be used as a biomedical material [17].

Figure 2 shows chitosan-based materials with different shapes and sizes: chitosan nanoparticles, chitosan nanofibers fabricated by electrostatic spinning technology, chitosan–pectin hydrogel grid scaffold prepared by 3D printing technology, chitosan core-alginate shell microspheres, chitosan-based fibers fabricated by solvent spinning technology, and 3D-printed chitosan porous structures [18].

## 3. Modification of Chitin and Chitosan

Although chitin and chitosan possess several useful qualities, their application is constrained by their poor solubility, small surface area, and porous makeup. To modify chitin and chitosan physically or chemically, various researchers have tested their theories. The two main benefits of modifying chitin and chitosan are to increase their solubility and improve their ability to absorb metals. The OH at the C3, C6, and NH2 at the C2 groups in chitosan undergo substitution reactions. These changes also improve the capacity of the membrane to swell in water [19,20,21].

In the molecular structure of chitosan, there are three types of active groups: amino groups, and primary and secondary hydroxyl groups at the C-3 and C-6 positions, which allow for chemical modification of chitosan. C6-OH is a main hydroxyl group with little steric hindrance, whereas C3-OH is a secondary hydroxyl group with a lot. As a result, the main hydroxyl group could freely rotate while the secondary hydroxyl group could not. The amino group is usually more reactive than the primary hydroxyl group, and the primary hydroxyl group is usually more reactive than the secondary hydroxyl group [22,23,24]. Chitosan can be chemically modified on the amino, hydroxyl, or both amino and hydroxyl groups to generate N-modified, O-modified, or N, O-modified chitosan derivatives [25] (Figure 3).

The proposed mechanism of chitosan modification can pass through one of the following four mechanisms: (a) free radical-induced conjugation to form a polyphenol chitosan conjugation; (b) carbodiimide chemical mechanism to form Schiff base compounds; (c) functional group conversion strategy, which converts the amino group of chitosan into an azide group, substituted carboxyl group, substituted mercapto group, and where the hydroxyl group can be azidate, aminated, oxidized to an aldehyde or carbonyl group, or further oxidized to a carboxyl group; or (d) conjugation of chitosan with polyphenol via enzymatic assisted coupling reaction, as shown in Figure 4a–d [26].

Chitosan has a low specific surface area (2–30 m^2^·g^−1^) [27] and is present in flake, which is unsuitable for many applications, so that it has been modified into beads to increase its value in different fields of application [19,20,21]. Due to the open structure and pores of chitosan, which has poor mechanical properties, shrinkage, and deformation in dry form and is only soluble in weak acids, numerous modifications have been made to it to improve its properties [28]. Chitosan contains amino and hydroxyl groups, so it is a poly-nucleophilic polymer. Nucleophilic substitution occurs. Protonation in NH_2_ groups results in the formation of NH_3_^+^ [29,30], such as N-alkylated chitosan prepared from Schiff base reactions followed by imine reduction by sodium borohydride. Furthermore, positive chitosan charges interacted with polyanion polymers such as alginate, carrageen, and pectin between the COOH and NH_3_^+^ groups [31]. Chitosan produce new functionalized derivatives via a grafting reaction [32]. The properties of grafted chitosan are determined by the side chains and the cross-liking agent [33]. Chitin threads are prepared to be used in absorbable suture fabrication, dressing, and biodegradable materials for human skin fiber growth [34]. 

Chitosan is a hydrophilic material used to impart hydrophilicity to some other polymer in its composites, such as polyacrylonitrile (PAN) [35]. Hydrophilic polymer nanoparticles were prepared by ionic gelation in mild conditions at room temperature, via two phases; one is for polymers such as chitosan polysaccharide (CS) and polyethylene oxide (PEO), and the other phase contains sodium tripolyphosphate (TPP). Calvo et al. prepared chitosan nanoparticles with high protein loading capacity, which were released within a week [36]. The morphology of these nanoparticles is spherical, as shown in Figure 5.

Nanoparticles of chitosan by using a variety of agents, the freeze-drying method increases shelf life. Chitosan nanoparticles of two different types were prepared by Alonso et al. in 1999 [37]. The prepared nanoparticles ranged in size from 300 to 400 nm, had a positive surface charge, and were efficient. These nanoparticles are used to address the nasal absorption of insulin. In addition, chitosan nanoparticles were used as a poly load of the anthracycline drug doxorubicin (DOX) [38,39]. In addition, these nanoparticles were used to load dextran sulphate to enhance its drug loading [40]. With cyclosporine serving as the model drug, chitosan nanoparticles (CSNP) were used to enhance the drug at the ocular surface. The size of these nanoparticles was 29 nm [41].

Chitosan-coated PLGA–Lecithin nanoparticles were prepared by the modified double emulsion method; these nanoparticles were used through the oral or nasal administration route. Figure 6 depicts the process of preparing CS–PLGA nanoparticles, and Table 1 summarizes the properties of the prepared nanoparticles [41]. 

Chitosan nanoparticles loaded by *Mycobacterium uaceae aerivatiuem* antineoplastic proteoglycans exhibit wide antimicrobial activity, as repeated by Tian and Groves (1999) [42]. The researchers prepared chitosan nanoparticles with particle sizes of 600–700 nm without using organic solvents and discovered that the two reactants affected the absorption and release of bovine, as well as that the initial Nanoparticles of chitosan by using a variety of agents, the freeze-drying method increases shelf life. Chitosan nanoparticles of two different types were prepared by Alonso et al. in 1999 release was followed by a steady release for 4 h in water [43].

Using self-aggregates of chitosan modified by deoxycholic acid, there is a novel and straightforward method of delivering adriamycin [44]. 

Deoxycholic acid and chitosan are covalently conjugated through an EDC-mediated reaction, resulting in self-aggregating chitosan nanoparticles. The Adriamycin active ingredient was physically trapped in nanoparticles, and the resulting self-aggregates were assessed using a variety of methods, including spectroscopy, which shows that these self-aggregates are spherical and that the drug concentration affects the shape of the particles [44]. 

These self-aggregates of chitosan modified by deoxycholic acid were used as DNA carriers by Kim et al. in 2001 [45]. Chitosan is used to overcome the side effects of drugs such as their solubility and hydrophobicity so that it can be used as a drug carrier. For example, commercial chitosan can be used to control body weight.

## 4. Chitosan as Biomaterial

Chitosan is a semi-crystalline polymer that is partially deacetylated chitin, and the degree of crystallinity is correlated with the degree of deacetylation. Chitin and fully deacetylated chitosan have the highest crystallinity, while intermediate levels of deacetylation have the lowest crystallinity [12]. Chitosan is stable, crystalline, and soluble in aqueous solutions at PH 7, but it is insoluble in weak acids due to the protonation of amino groups. Chitosan solutions could be extruded at higher PH values or in non-solvent baths, e.g., methyl alcohol. Chitosan polymer is used for industrial applications in such forms as fiber or film [46]. Chitosan solution’s cationic nature and high charge density make it potentially useful as a biomaterial. Chitosan forms soluble ionic complexes or complexes with water soluble anionic polymers such as alginates and synthetic polymers such as poly (acrylic acid) due to its charge density [21]. When used for local delivery of biologically active poly anions like GAGs and DNA, for example, ionic complexes release heparin to increase the efficacy of growth factor secreted by inflammatory cells [21,47,48,49,50,51]. 

Chitosan DNA complex is used to facilitate cellular transfection and prevent plasmid degradation by nucleases. Chitosan can become porous through freezing and lyophilization, making it useful for both tissue regeneration and cell transplantation [52]. Figure 7 illustrates how ice crystals form in the freezing process from a solution, grow from the ice crystal phase, and are then removed by lyophilization from a general porous mother. 

Pore size and pore orientation affect the mechanical properties of scaffold-based chitosan, with porous chitosan membranes having less elasticity (0.1–0.5 MPa) than non-porous membranes (5–7 MPa).

The maximum strain of the porous membrane of chitosan is greater than that of the non-porous membrane, and the 100% chemical modification of chitosan introduces new biological activity with modified mechanical properties. The chitosan NH_2_ group is reactive and capable of introducing side group attachment with several reactions that affect primarily crystallinity disruption with lower stiffness and alter soluble derivatives feasible in chemical reactions, such as alkyl derivatives of chitosan, which have lower solubility than chitosan itself and give aggregates miscible for c > 5. In addition, the basic properties of chitosan are hemostatic, cationic, and insoluble at pH 7, which are completely reversed by the solvation process to give ionic, water-soluble derivatives, and anticoagulant properties, so that chitosan can attack with unlimited side groups and be chosen according to specific needed functions, e.g., biological activity or modified properties, etc.

## 5. Applications of Chitosan and Chitosan Derivatives

Chitosan is a biodegradable, biocompatible, and non-toxic biomaterial; therefore, it has many applications in areas such as medicine, agriculture, food processing, cosmetics, and treatment of water. 

### 5.1. Agricultural Application

Chitosan can be used safely in agricultural applications because it does not pollute the environment or harm consumers; it is used as a leaf coating, fertilizer, and sea coating [10,53]. The use of chitosan in agricultural fields has increased exponentially, especially for germination improvement, leaf growth, retention of moisture, and fungal and disease reduction [53]. Chitosan boosts photosynthetic efficiency while enhancing plant tolerance to salinity, high temperatures, and drought [54]. Chitosan’s hydrophilic properties encourage water absorption while reducing transpiration [55]. To encourage plant development, chitosan can be employed as an additional carbon source in plant synthesis [56]. In comparison to the control group, seedlings treated with Cu-chitosan nanoparticles (NPs) at concentrations of 0.04 percent and 0.12 percent had greater rates of germination, seedling length, root length, and root number [57]. To encourage the mobilization of protein and starch to boost seedling growth, Cu-chitosan NPs can increase protease and -amylase activities [58].

### 5.2. Wastewater Treatment Applications

Chitosan contains both OH and NH_2_ groups as chelating agent groups, which allow it to be used in water treatments from wastes due to the high-power effect of these functional groups to bind with heavy dissolved metals present in wastewater such as Cu, Pb, Hg, and Ur [59]. Furthermore, chitosan can be used to break down food particles, particularly food proteins, as well as remove dyes from wastewater [60].

Cu (II) ion-containing wastewater was examined by Qin et al. [61] using sodium alginate and chitosan as treatment agents. They investigated how various parameters affected this ion removal. According to the findings, chitosan and sodium alginate worked better together than they did separately. Pesticide removal from wastewater was explored by Dwivedi et al. [61] using hydrogel beads made of chitosan and gold nanoparticles. The data obtained indicated that the synthetic sorbent has good pesticide removal capacity.

Heterogeneous catalysis is one technique that heavily relies on adsorption. Purification is one of the earliest documented uses of adsorption. Adsorbents are still used to clarify water [62]. Numerous adsorbents, including activated carbon, low-cost biomass adsorbents, waste sludge, rice husk, sugarcane bagasse, lignite, and chitosan, were used in the adsorption tests. Heavy metal contamination clean-up frequently employs chitosan. In order to study the adsorptions of three metal ions—Cu (II), Zn (II), and Pb (II) ions—in an aqueous solution, cross-linked chitosan was created by the homogeneous reaction of chitosan in an aqueous acetic acid solution with epichlorohydrin [63].

### 5.3. Food Industry Applications

Because of the high potential for toxicity of chitosan as a chelating agent and its high functional properties, as previously stated, chitosan is used in a variety of applications in the food industry, such as removing specific elements, particles, and undesirable materials, such as dyes and fats. In addition, it is widely used as a natural, safe preservative in the United States to store food [64,65].

There has been an increase in interest in recent years in studying the potential applications of chitosan as films or coatings in food packaging. This is due to their film-forming, antioxidant, and antibacterial properties, as well as their mechanical and barrier properties, which were studied as films. In order to extend the storability and shelf life of perishable commodities, these experiments sought to develop active packaging based on chitosan, either on its own or in combination with other materials. By combining chitosan with other natural antimicrobial agents, it is also possible to make food products that guarantee food safety against a variety of mutating and pathogenic bacteria [66,67,68].

An edible coating or thin edible film made of chitosan can be applied to food to act as active food packaging. The edible coating is a thin layer that is added to a food item and is generated by dipping the item in a chitosan solution or spraying, in which case, the film-forming solution is crushed up using an aerosol spray coating. Even though the chitosan film is a thin, prefabricated layer, once it is produced, it can be deposited on the surface of or in between food ingredients [66,67,69]. Studies using chitosan films and coatings on food products are shown in Table 2 [66].

### 5.4. Medical Applications

Chitosan is applicable in several medical industries, especially periodontal and orthopedic drug delivery, wound healing, and tissue engineering applications [86]. Surgical sutures, contact lenses, eye fluids, artificial skin, artificial blood vessels, bandages, sponges, burn dressings, blood cholesterol vessels, antitumor, antibacterial, antiviral, bane regenerator, antimicrobial, and hemostatic agents are the most well-known examples of these applications [86,87,88,89,90,91].

Chitosan inhibits tumor cell proliferation, and Liu et al. demonstrated that chitosan induces apoptosis in tumor cells by decreasing Bcl-2 and increasing Caspase-3 expression [92,93]. Carboxymethyl chitosan (CMCS) increases macrophage viability, deeply penetrates the tumor microenvironment, generates cytokines like TNF and IL-1, improves phagocytosis, and increases NO levels. Notably, CM-COS is not significantly toxic to normal liver cells but inhibits the growth of BEL-7402 and sarcoma cells in vivo [94,95]. Additionally, chitosan inhibits the invasion and metastasis of tumor cells. During tumor invasion and metastasis, matrix metalloenzymes (MMP) are involved in the breakdown of extracellular matrix, and MMP-2 can encode an enzyme that breaks down type IV collagen [96].

Chitosan is a great material for creating wound dressings because it can promote wound healing. It exhibits good antibacterial activity due to its alkaline amino groups, which cause the destruction of bacterial cells and protect the wound surface from microbial infection [97]. To improve the antibacterial and coagulation capabilities, Zhang et al. created a nanocomposite hydrogel utilizing zinc oxide (ZnO), chitosan, and aldehydic sodium alginate (SA) [98]. It significantly inhibited the growth of *Escherichia coli* and *Staphylococcus aureus*. I-sexual collagen hydrogels sulphated chitosan-doped by Shen et al. improved macrophage polarization from M1 to M2. The IL-4 and TGF-1 secreted by macrophages were stimulated, which resulted in an increase in collagen synthesis, regeneration epithelialization, and neovascularization [99].

Tissue engineering is an emerging interdisciplinary discipline combining material science, engineering mechanics, and biomedicine. The structure and function of damaged tissues and organs are repaired or replaced by cell transplantation combined with bioactive molecules and 3D scaffolds. Chitosan is similar to mucopolysaccharides of the extracellular matrix and is used as a scaffolding material for cartilage tissue engineering [100].

Kaviani et al. used the cryogel approach to construct nano porous scaffolds from chitosan, collagen, and nanohydroxyapatite, which lowered the rate of biodegradation, increased mechanical characteristics, and enabled cell proliferation and adhesion [58]. Porous scaffolds made from chitosan, gelatin, and silk proteins have higher compressive strength and modulus, whereas incidental chondrocytes can produce seed scaffolds and stimulate cartilage tissue regeneration [58]. By mixing gelatin, chitosan, and polyvinyl alcohol with nano-hydroxyapatite, Martino et al. created porous composite scaffolds [58]. Table 3 summarizes the biological application of chitosan and its derivatives. 

## 6. Electro-Spun Nanofibers: Process and Application

### 6.1. History of Electrospinning

In 1897, Rayleigh discovered electrospinning. The early 1700s saw the discovery of electrostatic effects on water behavior and their impact on the dielectric values of liquid excitation. Around the turn of the twentieth century, Cooley and Morton developed electrospinning technology. The rotatory electrode was then incorporated into electrospinning by Cooley. Formhals created yarns using electrospinning in 1930 without the use of a spinneret, and his technique and apparatus granted him a patent for his creation [119,120,121]. 

After that, Formhals submitted a patent for a different method of creating electrostatic polymer fibers: composite fibers made of several polymers. In 1969, Taylor conducted research on the composition of the polymer droplet formed at the needle tip by a strong electric field. This proved that the droplet assumed a cone-like shape, with jets emerging from vortices. The “Taylor cone” is the final name given to this cone. Additionally, factors affecting fiber stability, such as the electric field, flow rate, and experimental settings, were examined [122,123].

Compared to conventional spinning, electrospinning produces fibers at a much slower rate. Electrospinning produces yarn at a rate of 30 m per minute, compared to conventional spinning’s 200–1500 m per minute [119,120,124]. So, before 1990, melt spinning was the preferred method for creating fibers from natural and synthetic polymers, and only a small number of businesses were interested in electrospinning for fiber production. Nanometer-scale fibers cannot be produced through melt spinning [120,124].

For the purpose of removing harmful solvents and for use in tissue engineering applications, Dalton et al. applied an electro spun nanofiber web to tissue cells [125]. The use of electro-spun nanofibers today spans a wide range of potential uses [119,124,126]. Surface nanostructures may produce extraordinary phenomena, such as the lotus effect (self-cleaning effect). Since proteins, viruses, and bacteria all have dimensions in this range, the nanoscale is particularly important for biological systems. Electro spun fibers display a strikingly broad range of sizes when compared to the diameters of these things. Figure 8 depicts the dimensions of bacterial cells, proteins, viruses, and nanofibers [119,124,126,127].

### 6.2. Electrospinning Process

A high electric field (kV) is used to create micro or nanofibers from polymer solutions while maintaining ambient temperature and pressure. There are two primary setups for electrospinning devices: vertical and horizontal [119,124,128].

The major components of the electrospinning device are the power supply (high voltage), the syringe (spinneret), and the collector (electrode) [119,123,129].

High voltage creates electric charges on the surface of the polymer solution during the electrospinning process, and these charges build up on the surface. These charges possess repelling forces in a critical electric field that can dissipate the surface tension of the solution and the unstable charges. A Taylor cone-tip jet ejects a solvent-causing jet [119,123,124,130].

A stable jet created at the spinneret needle then changes into an unstable jet to create electro-spun fibers using a straightforward process. When the applied electric field reaches a critical value, jets shoot out of the cone tip, and the tensile force is transferred to the polymer, creating bending instability in that polymer. After that, a jet moves from the cone’s apex to a collector with opposing charges, which has the power to draw charged fibers to it. Jet travel causes the solvent to evaporate, leaving the dry fiber on the collector [119,122,123,124,131,132].

A typical electrospinning device can be created using a power supply with high voltage power, a syringe, and a collector electrode alone, as demonstrated in the experimental part. Polymeric materials can be electro spun to create continuous nanofibers, and there are a number of variables that affect the nanofibers’ properties. These variables are either processing variables (electric field) or polymer properties (concentration, viscosity, surface tension, and conductivity) (extrusion rate: distance from needle tip to collector) [119,123,124,129,133].

The definition of the electrospinning technique is the use of a strong electric field to spin polymer solutions into micro- to nanofibers at room temperature and atmospheric pressure (kV). Electrospinning devices can be set up in either a vertical or horizontal orientation [119,124,134,135,136], (Figure 9).

The three primary components of an electrospinning device are the power supply (high voltage), the syringe (spinneret), and the collector (electrode) [119,123,137,138]. When a polymer solution is subjected to a high voltage, electric charges accumulate on its surface. These charges repel one another so strongly that they can overcome the surface tension of the solution and form a Taylor cone in a critical electric field. When the electric field stretches the Taylor cone tip further, a charged jet is ejected. The jet eventually transforms into solid fibers due to solvent evaporation [123,139,140,141].

Using a simple process, a stable jet at the spinneret needle is converted to an unstable jet to generate electro-spun fibers. When the applied electric field reaches a critical magnitude, the surface tension is overcome by the charge repulsion force, and jets erupt from the cone tip, transferring the tensile force to the polymer and causing it to bend. The charged fibers are then attracted by jets, with opposing charges moving from the cone apex to the collector. As the solvent evaporates through the jet, it leaves dry fiber on the collector [119,124,131,139,141].

A high-voltage power source, a syringe, and a collector electrode may be used to construct a standard electrospinning apparatus that can be configured vertically or horizontally, as illustrated in Figure 7. Continuous nanofibers may be generated by electrospinning polymeric materials, and the quality of electro-spun nanofibers is determined by some criteria. Polymer physical properties (concentration, viscosity, surface tension, and conductivity) or processing factors (electric field, flow rate, needle tip to collector distance) are examples [119,124,139,141].

Prior to electrospinning, most polymers are dissolved in a range of solvents; when fully dissolved, they create a polymer solution. The polymer solution is then poured into the capillary tube in preparation for electrospinning. However, because some polymers emit unpleasant or even dangerous odors, the procedures should be done in well-ventilated areas [142]. In the electrospinning process, a polymer solution at the capillary tube’s end is exposed to an electric field, which causes an electric charge to form on the liquid’s surface. The repelling electrical forces outweigh the surface tension forces when the applied electric field is strong enough. The solvent evaporates, and a polymer is formed when a charged jet of the solution is eventually released from the Taylor cone’s tip. The jet is unstable and whips quickly in the area between the capillary tip and collector. Just past the spinneret’s tip, where the jet is stable, instability sets in. Consequently, the electrospinning technology makes the process of making a fiber simpler [143]. 

Solution parameters, process parameters, and environmental or “ambient” parameters are the three main groups into which factors that have an impact on the electrospinning process are divided [119,144]. These factors are gathered in order to produce smooth fibers without beads, so a thorough understanding of these factors is required in order to obtain electro-spun nanofibers that are bead-free.

### 6.3. Application of Electro-Spun Nanofibers

Nanofiber membranes are created using electrospinning and used in a variety of applications, including biomedicine, security, clothing, and nano-sensors. Nanofibers are used in the biomedical field to focus on tissue engineering, wound dressing, drug delivery systems, and enzyme immobilization because they resemble the majority of organs and tissues, including skin, collagen, cartilage, and bone [145]. 

Electro-spun nanofibers are unique in that they have a consistent morphology, a high surface area to volume ratio, and inter- and inner porosity. These characteristics make them promising as scaffold biomaterials [146,147]. Additionally, nanofibers improve protein absorption, cell growth, cell differentiation, and cell adhesion [148,149].

Due to their pores, nanofibers are also used in filtration as micro and nano filters based on membrane design and construction, allowing liquids and small particles to pass while arresting larger particle sizes (contaminants), similar to how paper coffee filters prevent undissolved particles from passing through their pores while allowing dissolved ones to do so [150,151,152,153].

Affinity membranes, which have numerous uses in the biomedical and environmental fields, were also developed to select immobilized targets and remove contaminant targets [146,147,152,153,154]. Although electro-spun nanofibers have many uses, including those for tissue engineering, drug delivery, enzyme immobilization, wound dressing, antibacterial properties, filtration, desalination, and protective clothing, the focus of this article will be on biomedical uses.

#### 6.3.1. Tissue Engineering Applications

Electro-spun nanofibers are used in tissue engineering scaffold construction [155]. The use of biodegradable and biocompatible nanofibers to provide target tissues has increased daily [156]. Due to the similarities between these fibers and the natural extracellular matrix, these fiber scaffolds had an impact on both cell-to-cell and cell-to-matrix communication and produced excellent growth factors [157]. 

#### 6.3.2. Drug Delivery Applications

Based on the observation that the drug’s rate of dissolution increased as the surface area of both the drug and the carrier increased, nanofibers were used to coat the drug and deliver it to the target site [158,159]. Anticancer medications, antibiotics, proteins, ribonucleic acid (RNA), and deoxyribonucleic acid are all delivered by electro-spun nanofibers (DNA) [160]. 

Recently, Yang et al. created a composite scaffold using nanofibers made of gelatin and polyvinyl alcohol (PVA) to transport raspberry ketone [161,162]. Additionally, to transport the growth factor calcium hydroxyapatite, Haider et al. prepared PLGA nanofibers [146,147].

#### 6.3.3. Enzymes Immobilization Applications

Immobilization of enzymes onto insoluble material is essential to improve durability and maintain the enzyme properties such as bioprocessing and long duration controls [163]. The immunized material essentials are biocompatible, durable and hydrophobic or hydrophilic [164].

Recently, electro-spun nanofiber prepared from the dual electrospinning process increased the enzyme immobilization [165]. Yet, there are some limitations that hinder this technique, and enzymes encapsulation and the enzyme immobilized on the fibers surface are limited [166], so that some chemical modification of the surface needs to overcome these limitations [167,168].

#### 6.3.4. Wound Dressing Applications

Inhibiting microorganisms, removing exudate, and protecting the wound site are all important functions of wound dressing. In addition to having antimicrobial properties, wound dressings provide a pleasant, moist environment to speed up the healing process [169,170,171]. Consequently, electro-spun wound dressings have more benefits than those made using traditional techniques [172]. These benefits include fiber pores, a large surface area, and the ability to stimulate fibroblast cells, making it suitable for use in cosmetic masks for skin cleansing and healing [142,173]. The electro spun nanofiber matrix incorporates various skin-treatment components [174].

Due to its capacity to create cationic clusters that can bind with anions on red blood cells, chitosan nanofibers have amazing hemostatic capabilities that can speed up platelet and red blood cell aggregation and ultimately reduce blood loss [175]. Additionally, this technique is successful even in individuals with coagulation abnormalities and is independent of the patient’s own clotting system [176]. By electrospinning, Ren and colleagues [177] created a medicinal dressing made of a composite of silk fibroin, chitosan, and halloysite nanotubes. Aluminum silicate-based halloysite nanotubes have a hollow tubular structure, can efficiently bind antibacterial medications, and can provide delayed, sustained drug release [177]. The findings showed that using halloysite nanotubes caused the loaded-drug release time to increase by about 8 days. Additionally, the electro-spun chitosan composite membrane demonstrated a better blood coagulation rate, improved tensile property, and antibacterial activity, all of which support its potential value as a medical dressing.

An anti-fibrinolytic medication known as tranexamic acid (TXA) is frequently used during trauma surgery and has been found to improve wound healing [178]. For hemorrhage control applications, Sasmal and colleagues [178] created TXA-loaded chitosan/PVA electro-spun nanofibers. The findings support the role of chitosan in hemostasis by demonstrating that the total blood clotting time of pure chitosan/PVA nanofibrous membranes decreased from 21,010 s to 1676 s as the amount of chitosan increased. Additionally, clotting time and plasma recalcification time were dramatically shortened after TXA was added to chitosan nanofibers, demonstrating the enormous potential of TXA-loaded chitosan nanofibers for managing civil and military hemostasis. Additionally, by fabricating chitosan within a hydrogel carrier template produced from cyclodextrin through proton exchange and complexation, Leonhardt and colleagues [179] observed the development of nanoscale features in chitosan mats. With nanofiber diameters of 9.23.7 nm and a macroscopic shape resembling a honeycomb, the assembled chitosan was highly entangled. When compared to commercially available absorbable hemostatic dressings, the chitosan-based composite hydrogels result in much less blood loss and faster timeframes for hemostasis.

#### 6.3.5. Antibacterial Applications

Many antibacterial hybrid electro-spun nanofiber scaffolds, including polyacrylonitrile/silver PAN and Ag nanofiber scaffolds, prepared by various research groups, have antibacterial activity against both gram-positive and gram-negative bacteria. Therefore, a wide variety of antimicrobial amidoxime was immobilized using PAN nanofibers. Amidoxime’s antibacterial action is caused by its binding to the Mg^2+^ and Ca^2+^ ions, which upset the balance of the bacteria and result in bacterial death [180]. 

### 6.4. Electrospinning of Chitosan

Template synthesis, drawing, phase separation, electrospinning, self-assembly, and other techniques have all been used to create nanofibers [181]. For the creation of micro- and nanofibers, electrospinning is one of these techniques that is particularly adaptable [182]. With their high surface area to volume ratio, oxygen permeable porosity, and variety of pore sizes, electro-spun nanofibers act as a wound dressing material by promoting fibroblast growth [183]. Chitosan has a low electro spinnability and needs a strong applied electric field to work properly because of its polycationic nature, which is brought on by the amino groups on its backbone [184,185]. A strong electric field is necessary because the polycationic nature of chitosan results in very viscous solutions with high surface tension. Additionally, the chitosan’s strong hydrogen bond network minimizes molecule exposure to the applied voltage [184,185]. 

Chitosan, like the majority of other polysaccharides, needs a strong acidic environment to dissolve properly, but this can be dangerous and prevent the use of chitosan fibers in some circumstances. Furthermore, when it comes to the creation of pure chitosan fibers, chitosan with a low molecular weight and low biological activity frequently produces better results. This has a negative impact on the biological functioning of the electro-spun chitosan [184,185,186]. As in all electrospinning techniques, changing parameters cause morphological changes in the spun fibers. The correlations between parameters and morphology are generally as follows: The diameter and length of the end fiber are decreased as the applied voltage is increased. Low chitosan concentrations can lead to fiber formation and breakdown, which increases fiber diameter and leads to morphological flaws [184,187].

Chitosan’s electro spinnability may be improved by using co-spinning polymers, either natural or synthetic. Frequently, chitosan is employed in the following fields: collagen, zein, silk fibroin, PEO, PVA, PLA, and zein [185,188]. Another method for resolving the issues with electrospinning chitosan is to chemically modify it to produce derivatives that are suitable for electrospinning, as with what was done with cellulose. More often than not, these derivatives have improved solubility and electro spinnability. Chitosan derivatives that have been investigated for this purpose include quarternized chitosan, hexanoyl chitosan, N-carboxyethyl chitosan, and others [189].

Chitosan is soluble in diluted aqueous formic, acetic, and lactic acids but insoluble in most mineral acidic media, alkalis, and water solutions. Chitosan dissolves when a small amount of acid is added to mixtures of water, ethanol, methanol, and acetone. Chitosan is a positively charged polyelectrolyte with a pH range of 2–6, which results in its greater solubility when compared to chitin. This characteristic makes chitosan solutions extremely viscous, making electrospinning difficult [190,191]. In addition, the three-dimensional network created by the potent hydrogen bonds prevents the polymeric chains from moving when they are exposed to an electrical field [192,193]. 

The unique properties of the polymer in solution, such as its polycationic nature, high molecular weight, and broad molecular weight dispersion, make chitosan electrospinning a challenging process. The inner tip–collector gap, the electric field voltage, the molecular weight, and the input velocity are just a few of the variables that have an impact on the quality of the electrospinning process and the finished product. When the electrostatic force in a solution is greater than the solution’s surface tension, the electrospinning process starts. When an electric field is applied, the surface of the polymer solution charges up. A strong electrical charge encourages jet extension and increases the volume of solution the needle can draw. On the other hand, a higher voltage leads to a longer stretch of the solution. This has a significant impact on electro-spun fiber morphology, frequently reducing their diameter, and raising the possibility of bead formation [190,194].

The feed rate, which regulates the number of solutions available, is another crucial component. A solution’s jet velocity and transfer rate are also impacted by its feed rate. For the evaporation of the solvent and the production of solid nanofibers, lower feed rates are preferred [119]. The solution must be removed from the tip at a rate that is substantially higher than the feed rate. Low feeding rates may prevent electrospinning, and high feeding rates may cause beaded large diameter fibers because sufficient solvent evaporation time must pass before the collector is reached [195]. 

SEM images of Figure 10 and Figure 11 show the effect of tip–collector distance and electrical field voltage, respectively, on the structure of electro-spun nanofibers. As shown in these figures, increasing gap distance and voltage not only decreases and refines nanofibers diameters, but also improves the quality of electro-spun nanofibers.

Another factor that affects the sizes and shapes of the nanofibers is the separation between the tip and the collector. In order to give the fibers enough time to dry before reaching the collector, a minimum distance is necessary; otherwise, beads have been observed when distances are either too close or too far apart [119]. This variable directly affects the strength of the electric field and the duration of the jet’s flight. The reduced tip–collector distance has a nearly identical effect on the circuit as increased voltage [196].

Additionally, rheological and electrical properties like viscosity, surface tension, conductivity, and dielectric strength are significantly influenced by molecular weight. High molecular weight nanofiber solutions produce fibers with a larger average diameter while too low a molecular weight solution tends to produce beads instead of fibers [197]. Polymer molecular weight, which is important to the electrospinning process, reveals how many polymer chains condense in a solution. It has been suggested that if there are sufficient interactions between molecules to compensate for chain entanglements, high molecular weights may not always be required for the electrospinning process. High molecular weight solutions typically result in the production of very long fibers. When a low molecular weight solution is employed, beads rather than fibers are produced [198]. 

### 6.5. Applications of Chitosan Nanofibers

#### 6.5.1. Chitosan Nanofibers in Tissue Engineering

Tissue-engineered scaffolds should be biocompatible and capable of mimicking the extracellular matrix to create a microenvironment conducive to cell growth. Natural bone extracellular matrix, for example, has a complex microstructure of multi-layered collagen fibers and calcium deposits with fiber sizes in the nano meter range [199].

Nanofibrous scaffolds, as nanostructured scaffolds, have the distinct characteristics of large surface area, high porosity, and mechanical strength, resulting in extraordinary biological properties such as mimicking the nanoscale properties of the extracellular matrix, promoting cell adhesion and migration, transporting nutrients, and discharging waste [200].

Among these properties are the functional groups of nanofibrous polymers, which can coordinate with other components to promote cell adhesion, proliferation, differentiation, and, eventually, tissue regeneration [201]. Nanofibrous scaffolds may be a viable solution for the synthesis of extracellular matrix substitutes with the required biological functions [201].

Chitosans are polysaccharide polymers that are similar to extracellular matrix components, so they can be metabolized and their degradation products can be stored as proteoglycans in vivo [202]. Because of their excellent biological properties, chitosan nanofibers are popular in tissue engineering research [202].

Nanofibrous scaffolds based on chitosan are widely used in a variety of applications, including nerve, bone, and cartilage engineering, cardiac and vascular tissue engineering, tendon, ligament, and skeletal muscle regeneration, and wound healing (Table 4). Because of their chemical properties, chitin and chitosan can be dissolved in solvents such as acetic acid, trifluoroacetic acid, formic acid, succinic acid, and 1,1,1,3,3,3-Hexafluoroisopropanol (HFIP) [203].

Electrospinning, self-polymerization, and thermally induced phase separation can all be used to create chitin/chitosan nanofibrous scaffolds [204]. Furthermore, chitin/chitosan nanofibers derived from polymer blending can alter the biological and mechanical properties of composite scaffolds [205]. To meet the requirements of tissue engineering, these composite nanofibrous scaffolds can mimic the nanoscale structure and porosity of the extracellular matrix [206,207].

**Table 4 polymers-15-02820-t004:** Tissue engineering applications of chitosan nanofibers.

Composition	Nanofiber Diameter	Cells	Application	Reference
Chitosan/montmorillonite/PVA nanofiber composite mesh	60–140 nm	Human dental pulp stem cells	Nerve tissue engineering	[208]
PVA/chitosan nanofiber composite scaffolds	94–410 nm	PC12 nerve cells	Nerve tissue engineering and repair	[209]
Polycaprolactone/chitosan nanofibers composite	110–240 nm	Schwann cell (SC)	Nerve tissue engineering	[210]
Chitosan/PVA/graphene oxide nanofibers composite	123–160 nm	ATDC5 cells	Cartilage tissue engineering	[211]
Chitosan/polyethylene oxide nanofiber composite	140 ± 41 nm	C2C12 myoblast cells	Tendon tissue engineering	[212]
Chitosan nanofiber scaffolds	50–450 nm	Primary ventricular cardiomyocytes	Cardiac tissue engineering	[213]
Polycaprolactone/chitosan nanofibers	150 ± 2 nm	Mouse model and sheep	Vascular tissue engineering	[214]
Chitosan/poly (vinyl alcohol) nanofibrous composite scaffold	137 nm	rMSC	Skeletal muscle regeneration	[215]
Chitosan nano-/micro fibrous double-layered composite	20–300 nm	Bovine chondrocytes	Cartilage tissue engineering	[216]

#### 6.5.2. Chitosan Nanofibers in Enzyme Immobilization

Nanofibers, in contrast to other nanomaterials, have recently attracted attention for enzyme immobilization not only due to their higher surface-area-to-volume ratio and greater enzyme-loading capacity, but also due to their higher chemical and mechanical protection, which is crucial to ensure physical resistance for the support. The disadvantages of nanoparticles, such as the behavior of aggregation, which affects the stability and activity of enzymes, are not present in nanofibers. They also do not need an additional procedure (such as centrifugation or membranes) to separate nano capsules. The gathered nanofibers form a macrostructure of nanofibers that is simple to separate and repurpose [217,218,219,220]. Therefore, chitosan has been used as a support for enzyme immobilization because of its benefits [221,222,223,224]. In fact, it is important to note that chitosan-based nanofibers should be further investigated as a support for enzyme immobilization due to the characteristics of chitosan combined with the benefits of nanofibrous structures. The use of chitosan-based nanofibers in enzyme immobilization is summarized in Table 5.

#### 6.5.3. Chitosan Nanofibers in Cancer Treatment

There are numerous anticancer medications, including Doxorubicin, Paclitaxel, Berberine, Methotrexate, Adriamycin, Curcumin, Vincristine, Mercaptopurine, Indomethacin, Ibuprofen, Ketoprofen, Dexamethasone, Tetracycline, Gemifloxacin, Tetanus Toxoid, and Folic acid. The quantity of these anticancer medications is loaded onto the carriers of fibers during in vitro and in vivo cancer treatment. The polycationic nature of chitosan makes it a good candidate in this field among other fiber carriers. Table 6 lists several anticancer drug-loaded chitosan fiber systems along with information on how they affected cancer tissues in vitro.

#### 6.5.4. Chitosan Nanofibers in Food Technology

Chitosan’s polycationic nature, biodegradability, nontoxicity, antimicrobial, chelating, mucoadhesive, and gelling properties set it apart. All these properties, combined with the high surface area: volume ratio of nanofibrous structures, make chitosan-based nanofibers suitable for a wide range of applications in food technology that have yet to be fully explored. As a result, the following sessions highlight some plausible studies in this field.

Food waste poses a problem for the food industry. According to the Food and Agriculture Organization of the United Nations [249], one-third of the world’s food is wasted, causing economic and environmental problems. Indeed, food packaging has emerged as a viable option for improving the quality and safety of food products by increasing their biological, physical, and chemical stability. It is necessary to avoid chemical contaminants, oxygen, microorganisms, light, and moisture to achieve these properties. As a result, active packaging with antimicrobial properties has received a lot of attention [250,251,252].

Nanofibers made of chitosan have enormous potential for use in active food packaging. It has been widely reported that chitosan is effective against bacteria, viruses, and fungi [253,254,255,256,257]. In addition, chitosan, a renewable and biodegradable polymer, presents an intriguing real alternative to petroleum-based polymers in the development of green packaging materials.

Arkoun et al. [258] prepared chitosan/poly(ethylene oxide) (PEO) and tested its antibacterial activity against pathogenic microorganisms such as *E. coli*, *Salmonella enterica serves Typhimurium*, *Staphylococcus aureus*, and *Listeria innocuous*.

The nanofibers had an irreversible antibacterial effect, resulting in a bactericidal rather than bacteriostatic mechanism, according to the authors. Furthermore, bacterial growth was reduced at pH 5.8, which is lower than the pKa of amine groups on chitosan, and as a result, the authors proposed that the nanofibers could be applied to foods such as yoghurt, milk, cheese, meat, and fish, where lactic acid is liberated during the storage period.

Cui et al. [259] formed chitosan/poly(ethylene oxide) loaded with tea tree oil and tested it against *Salmonella enteric* subsp. *enteric serovar Enteritidis* and *Salmonella typhimurium*, two food pathogenic microorganisms. When the concentration of tea tree liposome was increased to 50%, the tensile strength increased by around 350%. Tea tree liposomes improved the antibacterial effect of the chitosan/poly (ethylene oxide) nanofiber as well. Furthermore, the chitosan/poly (ethylene oxide) loaded with liposomes tea tree demonstrated a four-day stable antibiofilm activity in chicken meat samples.

To prevent microbial spoilage of fish fillets, Ceylan et al. [260] created electro spun chitosan/thymol/liquid smoke nanofibers. The electro spun chitosan/thymol/liquid smoke nanofibers effectively reduced nearly 60% of total mesophilic bacteria. Furthermore, the authors stated that the nanofibers were thermostable until 150 °C, which is within the temperature range used in traditional food preservation methods.

The field of food packaging is a developing one that will most likely expand in the coming years. There are still numerous opportunities to investigate the role of chitosan-based nanofibers in other pathogenic microorganisms and different food types. Furthermore, the effect of other biopolymers and bioactive compounds on food packaging properties, such as mechanical properties and water vapor permeability, can be evaluated. Chitosan-based nanofibers could also be used to control food quality by monitoring external and internal conditions.

The majority of bioactive substances, volatile molecules, antioxidants, and flavors are unstable or even degradable [261].

Therefore, in addition to increasing bioavailability, chitosan-based nanofibers could improve the stability of functional food compounds. Chitosan’s mucoadhesive property can be used to deliver bioactive molecules to the body in a manner similar to how drugs are delivered; the mucoadhesion of the functional compounds increases their absorption through the gastrointestinal tract [262,263]. These qualities have increased interest in using chitosan-based nanofibers as a drug delivery vehicle in addition to their non-toxic and biocompatibility benefits [264,265]. Although some research has been done using various nanomaterials as food carriers [266,267], chitosan-based nanofibers have not yet been sufficiently investigated in this regard.

As a delivery vehicle for curcumin, Shekarforoush et al. [268] created electro spun chitosan/xanthan gum nanofiber. The authors discovered that pH 2.2 had a lower release of curcumin from nanofibers than other studied pH values at 6.5 and 7.4. According to one theory, there were stronger electrostatic interactions between chitosan and xanthan at pH levels below the pKa of the amine groups on chitosan, such as pH 2.2, which decreased the swelling behavior of the nanofiber and, as a result, the diffusion of curcumin. As a result, curcumin was able to be transported using the electro spun chitosan/xanthan gum, which improved its stability and bioavailability.

Biocatalysts called enzymes are essential in the food industry. However, due to their poor operational stability, short shelf life, and challenging recovery and reuse, their green chemistry and substrate specificity are compromised [269,270]. These issues might be solved by the enzyme immobilization on chitosan-based nanofibers, though. Due to their high pore-interconnectivity, which improves mass transfer between the substrate and the enzyme and, as a result, the enzyme activity, they stand out as promising supports.

Chitosan’s functional groups can be used to functionalize surfaces, which can then be used to adsorb controlled-release enzymes. Lactose is hydrolyzed using chitosan/poly (vinyl alcohol) nanofibers. According to the authors, at 50 °C, the immobilized enzyme was more thermostable than the free enzyme. Additionally, after 28 days of storage at 4 °C and 25 °C, the immobilized -d-galactosidase retained 77% and 42%, respectively, of its activity. As a result, the thermal and storage stability was increased by immobilizing -d-galactosidase in chitosan/poly (vinyl alcohol) nanofibers.

Nanofibers made of chitosan have been developed for the immobilization of enzymes.

Indeed, they offer promising resources for the creation of hypoallergenic foods as well as a number of dairy products (baking, jam, jellies, wine, beer and juices). The application of chitosan-based nanofibers on the immobilization of enzymes in food processing, including amylase, trypsin, pectinase, protease, tyrosinase, lipase, pectin lyase, pectin, laccase, among many others, should therefore receive significant attention.

## 7. Limitations and Future Perspectives for Chitosan Application

The use of chitosan has several restrictions in addition to its many benefits. Chitosan’s low solubility at a neutral pH is its most significant drawback. Numerous chemical and physical procedures have been employed to increase its solubility in order to get around this drawback. Chitosan has three types of functional groups: an amino group at C-2, a primary hydroxyl group at C-6, and a secondary hydroxyl group at C-3. Recent research has attempted to modify the three reactive functional groups in order to improve antimicrobial properties as well as solubility. For example, adding CH_3_ to chitosan increased its solubility and allowed it to be used in a wider pH range. The addition of disaccharides and N-alkylation increased its solubility and antimicrobial activity against *E. coli* across a wider pH range. 

These findings collectively appear to suggest that chitosan can be modified in a variety of ways to increase its solubility and antimicrobial activity. Chitosan and its derivatives have received a lot of attention in recent decades due to their numerous applications in various fields. Several studies have shown that chitosan’s antimicrobial properties are affected by a variety of factors, including pH, temperature, Mw, metal chelation, and microorganism type. In vivo studies have also shown that chitosan and its derivatives can be used to treat microbial infections with no side effects. More research is needed, however, to determine the optimal chitosan conditions. 

Chitosan’s nontoxic, biocompatible, biodegradable, and antimicrobial properties indicate that this compound and its derivatives have a wide range of applications, which have been discussed in this review. In the future, chitosan may be used as an alternative to synthetic bactericides for crops because it has already been shown to be effective against the treatment of bacterial infections in animals. In the food industry, it can be used in food packaging materials to extend the shelf-life of food products, as well as in dressings that treat wounds in the pharmaceutical industry. 

However, more research is required to determine its mode of action. A more standardized and comprehensive description of procedures, on the other hand, is required to match the results of different investigators. More research is needed to understand the molecular events that underpin chitosan’s antimicrobial action. Finally, more research should be focused on improving chitosan’s antimicrobial activity while maintaining its low toxicity and biodegradability.

## 8. Conclusions

Chitosan-based polymers can be used in a variety of biological applications. Chitosan is a biomaterial that is both biocompatible and biodegradable. Chemistry increases the utility of chitosan. Chitosan and its derivatives are employed as nano-delivery systems in biotechnology. Electrospinning includes preparation, fiber configuration, material selection, intended applications, and the spinning method. Foundation polymers based on chitosan are anticipated to be used in biological applications more frequently.

The development of biodiversity materials with applications in biomedicine (tissue engineering, wound treatment, drug delivery), as well as environmental protection (air and water filters), is a promising field. Because of the difficulties, the most common method for electrospinning clean chitosan fibers is blending with a co-spinning agent, which has the advantages of easier electrospinning and complementary qualities for specific applications. The result is that chitosan has been combined with a wide range of synthetic polymers, such as poly (ethylene oxide), polyvinyl alcohol, poly (lactic acid), polycaprolactone, polyurethanes, polyamides, polyacrylates, polyethylene terephthalate, polyacrylonitrile, polyaniline, and natural polyproteins (collagen, gelatin, silk fibroin, sericin), as well as polyanionic polysaccharides (hybrid). The systematic analysis of the properties of these blend nanofibers revealed the advantages and disadvantages of each method, the main rules that should be followed when aiming for a specific morphology, and the impact of the co-spinning agents on the fiber properties, which further directs their use. All this foundational knowledge in chitosan electrospinning is useful for the design of materials for real-world applications, which appear to be focused on the fabrication of complex chitosan-based nanofiber blends or composites in order to meet the need for multifunctionality.

## Figures and Tables

**Figure 1 polymers-15-02820-f001:**
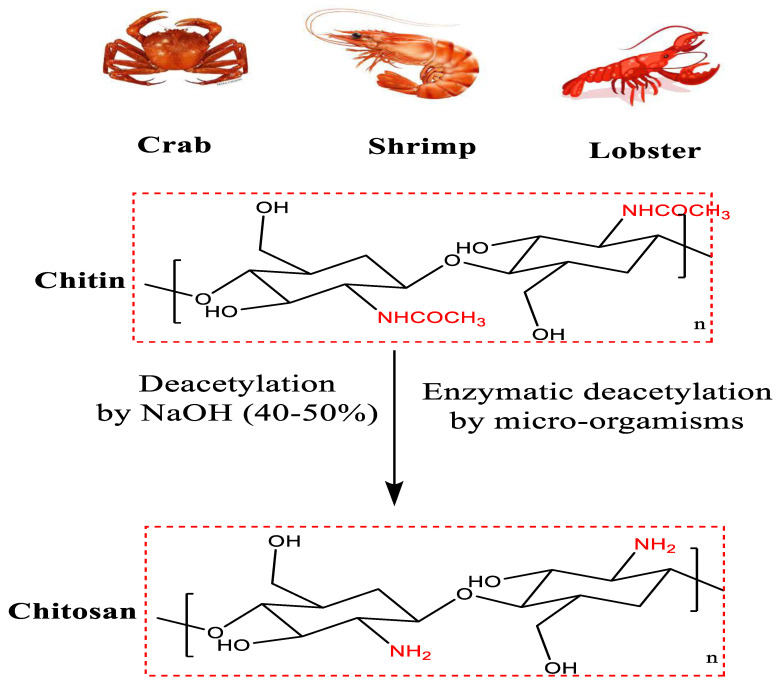
Chemical Structure and some sources of chitin and chitosan [5].

**Figure 2 polymers-15-02820-f002:**
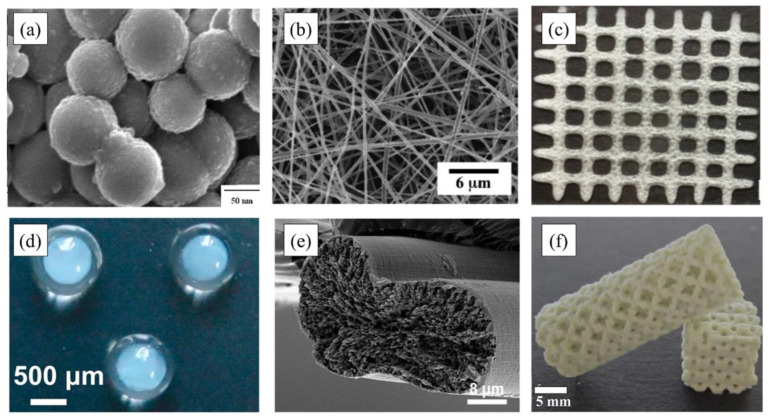
Chitosan-based materials with different shapes and sizes: (**a**) chitosan nanoparticles; (**b**) chitosan nanofibers fabricated by electrostatic spinning technology; (**c**) chitosan–pectin hydrogel grid scaffold prepared by 3D printing technology; (**d**) chitosan core–alginate shell microspheres (**e**) chitosan-based fibers fabricated by solvent spinning technology; and (**f**) 3D-printed chitosan porous structures. Copyright 2023. Reproduced with permission from Elsevier Science Ltd. [18].

**Figure 3 polymers-15-02820-f003:**
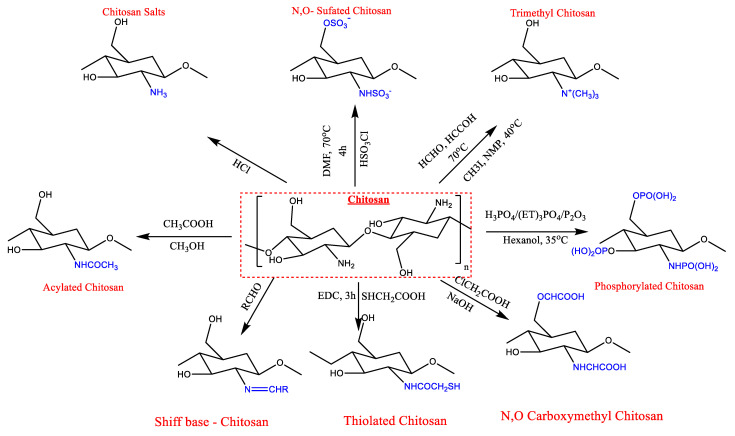
Chemical modification of chitosan.

**Figure 4 polymers-15-02820-f004:**
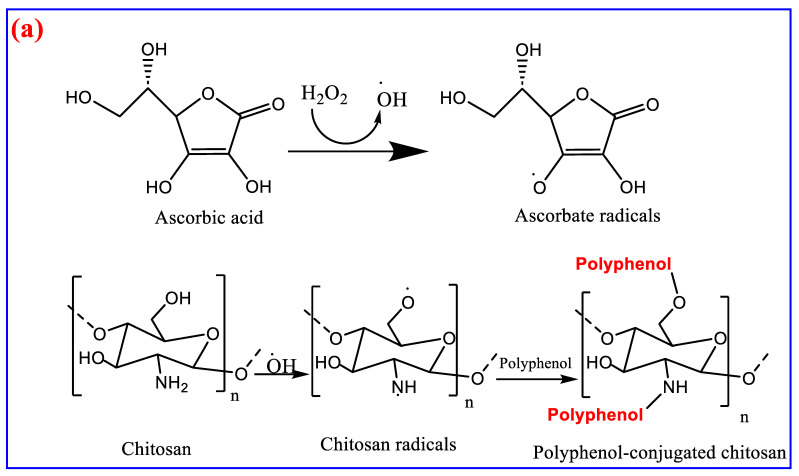
Proposed mechanisms (pathway) for chemical modification of chitosan via conjugation (**a**) free radical-induced conjugation to form a polyphenol chitosan conjugation; (**b**) carbodiimide chemical mechanism to form Schiff base compounds; (**c**) functional group conversion strategy, or (**d**) conjugation of chitosan with polyphenol via enzymatic assisted coupling reaction.

**Figure 5 polymers-15-02820-f005:**
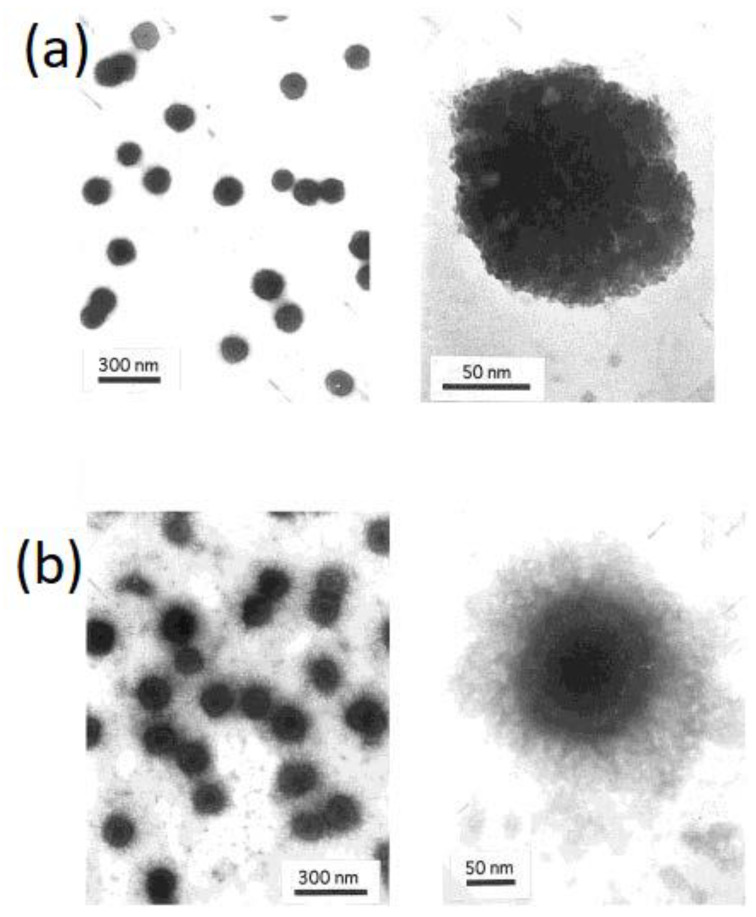
Electron transmission microphotography of: (**a**) CS nanoparticles; (**b**) CS/PEO–PPO nanoparticles (concentration of PEO–PPO in the chitosan solution: 10 mg/mL) prepared by Calvo et al. [36], reproduced with permission from John Wiley and Sons.

**Figure 6 polymers-15-02820-f006:**
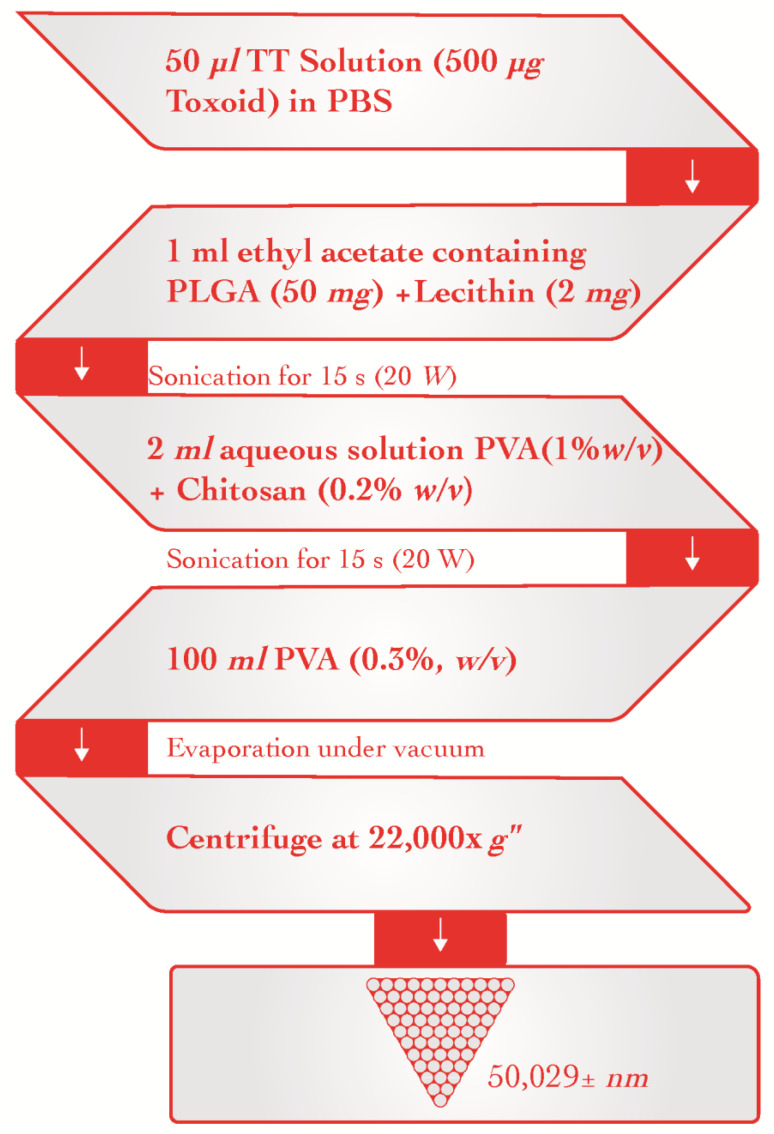
Flowchart showing the creation of chitosan–PLGA particles step-by-step.

**Figure 7 polymers-15-02820-f007:**
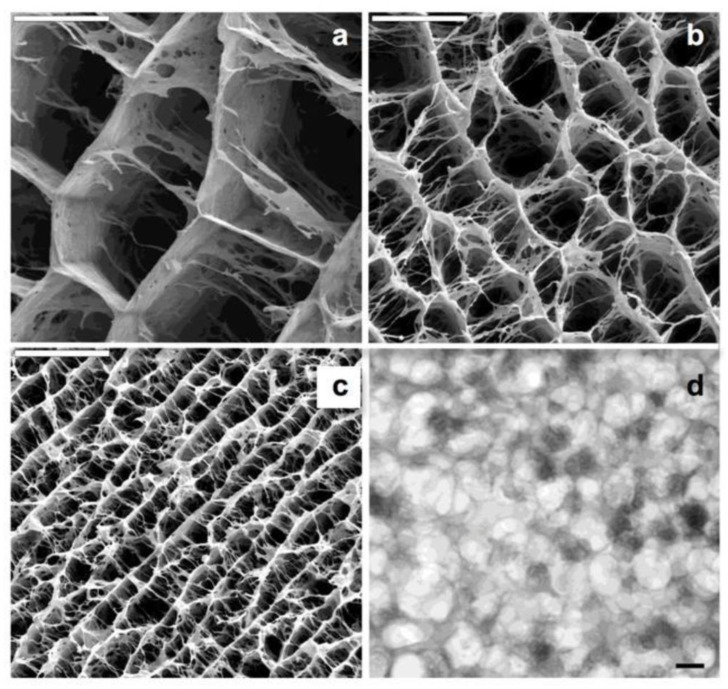
SEM micrographs of different hybrid hierarchical structures resulting from freezing hydrogel nanocomposites, with identical CHI and calcium phosphate composition (93.25 and 6.75 wt.%, respectively) at different rates of freezing: (**a**) 0.7 mm/min; (**b**) 2.7 mm/min, and (**c**) 5.7 mm/min. Scale bars are 50 μm. TEM (**d**) micrographs of ACP nanoclusters forming the ACP/CHI hierarchical structure. Ref. [52] Reproduced with permission from MDPI.

**Figure 8 polymers-15-02820-f008:**
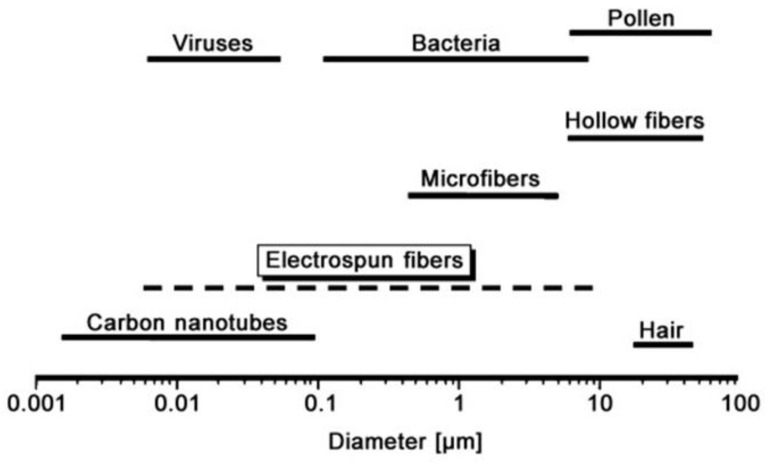
Position of nanofibers between protein, bacteria, and viruses [126]. reproduced with permission from John Wiley and Sons.

**Figure 9 polymers-15-02820-f009:**
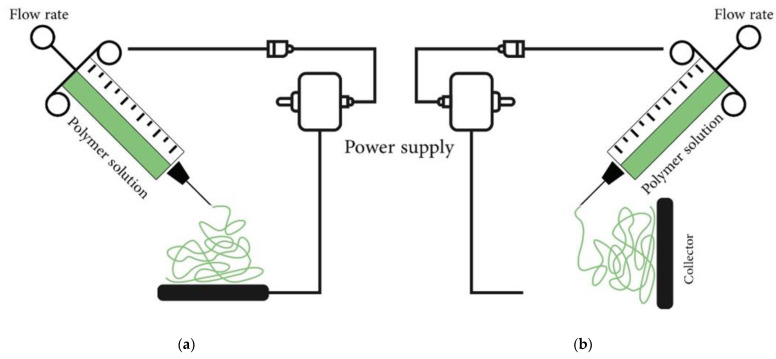
A diagram of electrospinning equipment with a syringe injecting polymer solution into an electric field generated by a high voltage power source between the spinneret and the grounded collector; a high voltage is applied. (**a**) A typical vertical arrangement of an electrospinning apparatus. (**b**) A typical horizontal setup of an electrospinning apparatus [119].

**Figure 10 polymers-15-02820-f010:**
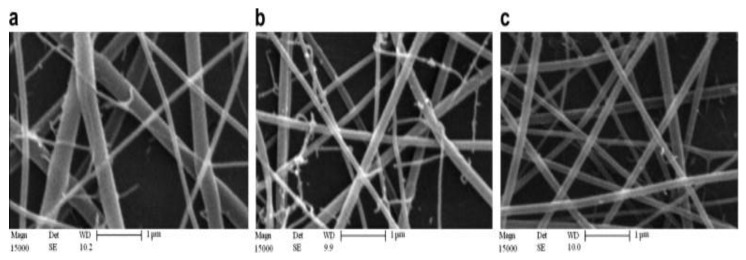
SEM images of 5 wt.% chitosan hydrolyzed 48 h nanofibers in aqueous acetic acid 90%, tip needle–collector distance: 14 cm (**a**), 15 cm (**b**), 16 cm (**c**). [119] Copyright 2023. Reproduced with permission from Elsevier Science Ltd.

**Figure 11 polymers-15-02820-f011:**
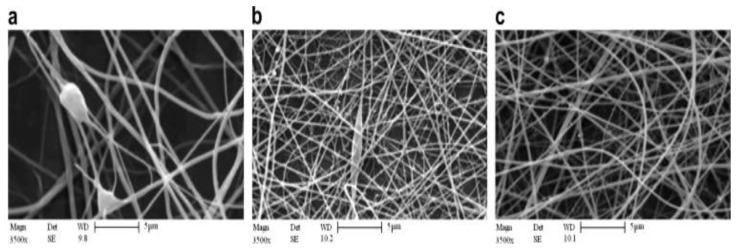
SEM images of 5 wt.% chitosan hydrolyzed 48 h nanofibers in aqueous acetic acid 90%, electric field voltage: 14 kV (**a**), 16 kV (**b**), 17 kV (**c**). [196] Copyright 2023. Reproduced with permission from Elsevier Science Ltd.

**Table 1 polymers-15-02820-t001:** Particle size zeta potential theoretical loading, and encapsulation efficiency values of CS [41].

Polymer	Protein Loaded	Size (nm)	Potential (mV)	Theoretical Lading (%)	Encapsulation Efficiency (%)
PLA	Tetanus Toxoid	192 ± 12	−47.9 ± 1.5	1	36.7 ± 0.3
PEG-PLA	Tetanus Toxoid	196 ± 20	−23.9 ± 1.2	1	31.1 ± 0.5
CS-PLGA	Tetanus Toxoid	500 ± 29	−21.8 ± 1.1	1	90.0 ± 3.6
CS	Tetanus Toxoid	354 ± 27	−37.1 ± 5.9	10	55.1 ± 3.4
CS	Insulin	337 ± 14	−36.9 ± 0.3	40	94.7 ± 2.1

**Table 2 polymers-15-02820-t002:** Chitosan-based films have been used in a wide variety of food products.

Chitosan Film/Coated	Food	References
Chitosan	Citrus fruit	[70,71]
Chitosan	Logan fruit	[72]
Chitosan	Green coffee beans	[73]
Chitosan	Frozen salmon	[74]
Chitosan	Tankan citrus fruit	[75]
Chitosan	Apples	[76,77]
Chitosan	Mushroom	[78]
Chitosan	Mangoes	[70,71,79]
Chitosan	Litchi fruit	[80,81]
Chitosan	Salmon fillets	[82]
Chitosan	Tomatoes	[83]
Chitosan	Strawberries	[83]
Chitosan	Peach fruit	[72]
Chitosan	Fresh-cut Chinese water chestnut L.T.	[84]
Chitosan	Silver carp	[85]

**Table 3 polymers-15-02820-t003:** Biological applications of chitosan and its derivatives.

Composite/Properties	Stimulatory Effects	References
Environmental purposes		
Flocculation	Removes a variety of contaminants from wastewater in an efficient manner	[101]
Metals and organic compounds adsorption	Removes pathogens, radioactive materials, heavy metals, colors, organic chemicals, and inorganic nutrients (nitrates and phosphates)	[101]
Agricultural purposes		
Biocontrol agent	Safe alternative to the use of pesticides and agrochemicals as a biocontrol material against many pathogenic microorganisms	[102]
Enhance crop production	- Effectively raises the productivity of various agricultural plants - Effectively encourages plant development in a variety of crops	[103]
Aquafeed additives	Positively impacts the growth, digestive enzymes, body composition, intestinal bacterial count, immunological response, and hematological and liver health of commercial freshwater fish	[104,105]
Biomedical purposes		
Chitosan microspheres	- Used as a carrier for targeted and prolonged delivery of drugs - Increases the bioavailability of degradable substances such as protein - Increases uptake of hydrophilic substances across the epithelial layers	[106]
Chitosan mesh membrane	Decreases wound healing time and increases the recovery of the granular layer	[107]
Chitosan collagen blend membrane	Increases the antibacterial activity against *E. coli* and *S. aureus* and decreases the excessive dehydration of the wound	[108]
Alginate/carboxymethyl chitosan blend fibers	Increased water-retention and increased antibacterial activity against *S. aureus*	[109]
Composite nanofibrous membranes (NFM) of collagen and chitosan	Increased wound-healing and increased tissue regeneration	[110]
Electro spun chitosan fiber with polyethylene oxide	Used effectively as surface layers on the wound site in periodontal disease	[111]
Chitosan membranes loaded with Tetracycline hydrochloride or silver sulfadiazine	Increased wettability, decreased swelling rate, water vapor permeability, and tensile strength, and increased antimicrobial activity against *E. coli* and *S. aureus*	[112]
Chitosan titanium dioxide composite membranes	Increased antimicrobial activity against *S. aureus*, decreased oxidative stress and apoptosis of fibroblast cell sand, increased proliferation in L929 fibroblast cells	[113]
Chitosan nano silver dressing	Increased wound-healing using the non-invasive dressing	[114]
Chitosan sponges loaded with norfloxacin	The dressing can conduct the role of normal skin and the antibiotic release is swelling-controlled	[115]
Chitosan-gelatin sponge	Increased antimicrobial activity against *E. coli* K88 over penicillin Increased antimicrobial activity against *S. aureus* over cefradine	[116]
Photo cross linkable chitosan hydrogel containing fibroblast growth factor-2	Increased wound healing in diabetic and normal mice	[117]
Carboxymethyl chitosan alginate hydrogel	Increases Bactericidal properties toward *S. aureus* and *E. coli*, and antibiotic continues to be released from the hydrogel	[118]

**Table 5 polymers-15-02820-t005:** Chitosan–based nanofibers in enzyme immobilization.

Composition	Component’s Ratio	Enzyme	References
Chitosan/poly (vinyl alcohol) composite in 0.1 M Sodium acetate	1:5,4	Phytase by entrapment	[225]
Chitosan/poly (vinyl alcohol) composite in 1% acetic acid	1:1	B-d-galactosidase by entrapment	[226]
Chitosan/poly (vinyl alcohol) composite in water	1:6	Urease by Cross-linking	[227]
Chitosan/poly (vinyl alcohol) composite in 0.5% acetic acid	1:10	Lysozyme by Cross-linking	[228]
Chitosan/poly (vinyl alcohol) composite in 2% acetic acid	1:4	Laccase by Cross-linking	[229]
Chitosan/polyethylene oxide composite in 1% acetic acid	22:1	Trypsin by Adsorption/Covalent	[230]
Chitosan/poly (ethylene oxide) composite in 90% acetic acid	95:5	Glucose oxidase by Cross-linking	[219]
Chitosan/polyamide 6 composite in Formic acid and Acetic acid (2:1 *v*/*v*)	1:9	Laccase by Cross-linking	[231]
Chitosan/Cellulose monoacetate composite in acetone 99%	1:5, 2:5, 1:1, 7:5	Protease by Adsorption/Cross linking	[232]
Chitosan/gelatin composite in 60% acetic acid	4:6	Peroxidase by Cross-linking	[233]

**Table 6 polymers-15-02820-t006:** The chitosan nanofibers as anticancer drug delivery in vitro conditions.

Composition	Anticancer Drug	Cancer Type	References
Chitosan	Fe_3_O_4_/under magnetic field	HFL1	[234]
Chitosan	Fe^2+^/under magnetic field	HFL1	[234]
Chitosan	Glutaraldehyde/under magnetic field	HFL1	[234]
Chitosan/poly(ε-caprolactone) composite	5-Fluorouracil	B16F10	[235]
Chitosan/polyvinyl alcohol-g-C_3_N_4_-g-C_3_N_4_ composite	5-Fluorouracil, Doxorubicin, Paclitaxel	MCF-7	[236]
Chitosan/polycaprolactone composite	Resveratrol, ferulic acid	HaCat, A431	[237]
Chitosan/polyethylene oxide composite	Berberine	HeLa, BT474, MCF-7, MDA-MB-468	[237]
Polyethylene oxide/chitosan/graphene oxide composite	Doxorubicin	A549	[238]
Chitosan/gelatin composite	Resveratrol	HT29	[239]
Polyvinyl alcohol/chitosan/Au composite	Doxorubicin	SKOV3	[240]
Polycaprolactone/chitosan composite	Cisplatin	Erlich ascites carcinoma	[241]
Chitosan/polyethylene oxide/hyaluronic acid composite	Paclitaxel	DU145	[242]
Chitosan/cobalt ferrite/TiO_2_ composite	Doxorubicin	B16F10	[234]
Chitosan/poly (lactic acid)/TiO_2_/graphene oxide	Doxorubicin	A549	[243]
Chitosan/PVA/graphene oxide/Si composite	Curcumin	MCF-7	[244]
Poly (ε-caprolactone diol)/polyurethane/chitosan/Au TiO_2_	Temozolomide	U-87 MG	[245]
Graphene oxide/chitosan composite	Curcumin	MCF-7, HepG2, L929	[244]
Poly (lactic-co-glycolic acid)/chitosan/SiO_2_ composite	Doxorubicin	HeLa	[246]
Chitosan/polyvinyl alcohol/MoS_2_ composite	Doxorubicin	HT29, HT29 cell lines	[247]
Chitosan	Cupric oxide	A549	[248]

## Data Availability

Data is contained within the article.

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
