# Peer review of "A Review of Chitosan and Chitosan Nanofiber: Preparation, Characterization, and Its Potential Applications"

_polymers, 2023, doi:10.3390/polym15132820_

Round 1

Reviewer 1 Report (New Reviewer)

Author Ibrahim et al. describe "A review of chitosan and chitosan nanofiber: preparation, characterization and its potential applications".

Following are my comments for the editorial decision;

1: Need to make a Table related to the biological application of chitosan (with different sizes) and its several derivatives. Provide the name of the composite in Tables 3, 4, and 5. The active concentrations and mechanisms also need to be considered. Although the author made some tables mechanisms are not in detail.

2: Some figures have poor resolution, like Figures 6 and 7.

3: Provide chemistry (reaction) about the preparation of chitosan and derivatives. And also for the making of some conjugates of chitosan.

4: Provide a section about the limitation and future perspectives for applying chitosan and derivatives.

5: Provide some images about the composites, conjugates or nanoparticles. nanofibres.

Please check the typo error, font size, and spelling.

Author Response

Manuscript title: A review of chitosan and chitosan nanofiber: preparation, characterization and its potential applications

Manuscript ID:   

polymers-2390995-R1

            Authors would like to express their appreciation for the extremely thoughtful Editor and reviews. We have been able to revise and improve the manuscript as a result of reviewer’s valuable feedback.

Reviewer 1

Author Ibrahim et al. describe "A review of chitosan and chitosan nanofiber: preparation, characterization and its potential applications".

Following are my comments for the editorial decision;

1: Need to make a Table related to the biological application of chitosan and its several derivatives.

Table 3 related to the biological application of chitosan and its several derivatives has been provided in the revised manuscript as reviewer comment

 Provide the name of the composite in Tables 3, 4, and 5. The active concentrations and mechanisms also need to be considered.

It has been corrected in the revised manuscript as reviewer comment

2: Some figures have poor resolution, like Figures 6 and 7.

Figures have been modified in the revised manuscript as reviewer comment

3: Provide chemistry (reaction) about the preparation of chitosan and derivatives. And also, for the making of some conjugates of chitosan.

It has been corrected in the revised manuscript as reviewer comment

4: Provide a section about the limitation and future perspectives for applying chitosan and derivatives.

Limitations and future perspectives for chitosan application added as separate section before conclusion in the revised manuscript as reviewer comment

5: Provide some images about the composites, conjugates or nanoparticles. nanofibers.

Images have been added in the in the revised manuscript as reviewer comment

Comments on the Quality of English Language: Please check the typo error, font size, and spelling.

Typo error, font size, and spelling have been checked and corrected in the revised manuscript as reviewer comment

Reviewer 2 Report (New Reviewer)

The work of Ibrahim et al. entitled “A review of chitosan and chitosan nanofiber: preparation, characterization and its potential applications” is bibliographic review based on the used of chitosan for the preparation of nanofibers and their corresponding applications. The topic is pertinent, but it is not a well written manuscript. The different sections that the manuscript deals with are exposed in a very disorderly and repetitive manner throughout the entire text. The images are of poor quality, and any image of a fiber containing chitosan is also missing. Although some interesting information has been presented, the manuscript needs a deep restructuring, the duplication of information. I would recommend a rejection since it does not meet journal standards.

Regarding the quality of English, it can be improved, there are many repetitions and spelling errors.

Author Response

Manuscript title: A review of chitosan and chitosan nanofiber: preparation, characterization and its potential applications

Manuscript ID:   

polymers-2390995-R1

            Authors would like to express their appreciation for the extremely thoughtful Editor and reviews. We have been able to revise and improve the manuscript as a result of reviewer’s valuable feedback.

Reviewer 2

Comments and Suggestions for Authors

The work of Ibrahim et al. entitled “A review of chitosan and chitosan nanofiber: preparation, characterization and its potential applications” is bibliographic review based on the used of chitosan for the preparation of nanofibers and their corresponding applications. The topic is pertinent, but it is not a well written manuscript. The different sections that the manuscript deals with are exposed in a very disorderly and repetitive manner throughout the entire text. The images are of poor quality, and any image of a fiber containing chitosan is also missing. Although some interesting information has been presented, the manuscript needs a deep restructuring, the duplication of information. I would recommend a rejection since it does not meet journal standards.

The manuscript revised carefully and improved. In addition, new sections added to enhance the manuscript quality.  The images revised and replaced with another higher resolution one. image of a fiber containing chitosan have been added in the revised version. Overall, the whole manuscript revised as reviewer comment

Comments on the Quality of English Language: Regarding the quality of English, it can be improved, there are many repetitions and spelling errors.

Typo error, font size, and spelling have been checked and corrected in the revised manuscript as reviewer comment

Reviewer 3 Report (New Reviewer)

The article “A review of chitosan and chitosan nanofiber: preparation, characterization and its potential applications” authored by Marwan A. Ibrahim, Mona H. Alhalafi, El-Amir M. Emam, Hassan Ibrahim and and Rehab M. Mosaad present some aspects related to the preparation of chitosan and its applications.

Observations

- Maybe it is worth to delete chitin from the title because the article is focused mainly on chitosan nanofibers applications.

-The language must be improved. Ex: pag. 16, chapter 6.4.3: It is crucial to immobilize enzymes on insoluble material to increase their durability and maintain their properties for bioprocessing and long-term controls. I suggest to rephrase the sentence: It is crucial to immobilize enzymes on insoluble material to increase their stability and performance of biocatalytic processes. The term "durability" is not correctly used.

Other example: Recently, enzyme immobilization was increased by electrospun nanofiber made using a dual electrospinning process: Recently, enzyme immobilization efficiency was increased by use of nanofibers obtained by a dual electrospinning process.

-Chapters 6 and 7 could be compiled to avoid confusion.  Chapter 6 presents the electrospinning process, some of the applications referring directly to chitosan. In the Chapter 7 “Electrospinning of chitosan" the authors discuss the factors that influence the electrospinning process. Among others, the author states that "The molecular weight of the chitosan has a significant effect on how it functions in terms of electrical properties like viscosity, surface tension, conductivity, and dielectric strength.[173]". Viscosity it is not an electrical property!

Next, other applications of chitosan nanofibers are presented.

- The figures quality must be improved.

This article can not be published in the current form.

-

Author Response

Manuscript title: A review of chitosan and chitosan nanofiber: preparation, characterization and its potential applications

Manuscript ID:   

polymers-2390995-R1

            Authors would like to express their appreciation for the extremely thoughtful Editor and reviews. We have been able to revise and improve the manuscript as a result of reviewer’s valuable feedback.

Reviewer 3

Comments and Suggestions for Authors

The article “A review of chitosan and chitosan nanofiber: preparation, characterization and its potential applications” authored by Marwan A. Ibrahim, Mona H. Alhalafi, El-Amir M. Emam, Hassan Ibrahim and Rehab M. Mosaad present some aspects related to the preparation of chitosan and its applications.

Observations

  1. Maybe it is worth to delete chitin from the title because the article is focused mainly on chitosan nanofibers applications.

The title doesn’t contain chitin in the title. the revised manuscript title is: A review of chitosan and chitosan nanofiber: preparation, characterization and its potential applications

  1. The language must be improved. Ex: pag. 16, chapter 6.4.3: It is crucial to immobilize enzymes on insoluble material to increase their durability and maintain their properties for bioprocessing and long-term controls. I suggest to rephrase the sentence: It is crucial to immobilize enzymes on insoluble material to increase their stability and performanceof biocatalytic processes. The term "durability" is not correctly used.

It has been rephrased and revised in the revised manuscript as reviewer comment

  1. Other example: Recently, enzyme immobilization was increased by electrospun nanofiber made using a dual electrospinning process: Recently, enzyme immobilization efficiency was increased by use of nanofibers obtained by a dual electrospinning process.

It has been rephrased and revised in the revised manuscript as reviewer comment.

  1. Chapters 6 and 7 could be compiled to avoid confusion.  Chapter 6 presents the electrospinning process, some of the applications referring directly to chitosan.

Chapters 6 and 7 were compiled in one chapter in the revised manuscript as reviewer comment.

  1. In the Chapter 7 “Electrospinning of chitosan" the authors discuss the factors that influence the electrospinning process. Among others, the author states that "The molecular weight of the chitosan has a significant effect on how it functions in terms of electrical properties like viscosity, surface tension, conductivity, and dielectric strength.[173]". Viscosity it is not an electrical property!

It has been checked and corrected in the revised manuscript as reviewer comment.

  1. Next, other applications of chitosan nanofibers are presented.

It has been stated in the revised manuscript as reviewer comment.

  1. The figures quality must be improved.

Its quality has been improved checked and corrected in the revised manuscript as reviewer comment.

This article cannot be published in the current form.

The manuscript has been revised as reviewer comments in the revised submitted form

Reviewer 4 Report (New Reviewer)

Dear authors, 

Thanks for the interesting review.

The general opinion about the article is very contradictory. The formatting of the article is on a low level. It is necessary to bring the appearance of the article to a general view. It is necessary to clearly follow the template of the magazine, at least in the tables of contents and subsections. It is difficult to read in some places. Include some formal sections at the end of the article. Here are some comments I could find:

1. Section two has the first two subsections numbered 1.1.2.1 and 1.1.2.2. Maybe I do not understand the numbering system, but in the text this numbering looks strange, because the section itself has the number 2. Then there are no subsections 1.1 or 1.1.1. So I suggest that we review the numbering of the subsections here and below to get on the same view.

2. Line 70: Water and the majority of organic solvents can dissolve chitin, a hydrophobic substance. Check this sentence. Can chitin dissolve in water? Also, the sentence is difficult to understand. 

3. Check the formatting of the captions to the pictures. It seems to me to be different in the different pictures. 

4. Line 135: what is meant by anti-biological activity? In my opinion, we need to somehow decipher this concept. Does it mean that it kills all living things in general?

5. Check the formatting throughout the text - some references are highlighted and some are not. Also check against the sample citation style: "*.[*]." Is a period before the square brackets necessary?

6. Line 328: Surface nanostructures may produce extraordinary phenomena, such as the lotus effect (self-cleaning effect). Remove word repetition. 

7. "A high-voltage power source, a syringe, and a collector electrode may be used to construct a standard electrospinning apparatus that can be configured vertically or horizontally as illustrated in Fig. 6." Maybe Figure 7, not 6?

8. Why is section 8 in red font? You should have carefully reviewed and formatted the article before submitting it to the journal. 

9. Among these properties are the functional groups of nanofibrous polymers, which can coordinate with other components to promote cell adhesion, proliferation, differentiation, and, eventually, tissue regeneration. (Sofi, Ashraf, Beigh, & Sheikh, 2018). Reference in a wrong format. 

10. After section 8.4.1 comes section 7.2 Why so?

11. Conclusions section: 5. Conclusion. Why 5? 

12. Section 6. References. Redo the numbers.

Author Response

Manuscript title: A review of chitosan and chitosan nanofiber: preparation, characterization and its potential applications

Manuscript ID:   

polymers-2390995-R1

            Authors would like to express their appreciation for the extremely thoughtful Editor and reviews. We have been able to revise and improve the manuscript as a result of reviewer’s valuable feedback.

Reviewer 4

Comments and Suggestions for Authors

Dear authors, 

Thanks for the interesting review.

The general opinion about the article is very contradictory. The formatting of the article is on a low level. It is necessary to bring the appearance of the article to a general view. It is necessary to clearly follow the template of the magazine, at least in the tables of contents and subsections. It is difficult to read in some places. Include some formal sections at the end of the article. Here are some comments I could find:

  1. Section two has the first two subsections numbered 1.1.2.1 and 1.1.2.2. Maybe I do not understand the numbering system, but in the text this numbering looks strange, because the section itself has the number 2. Then there are no subsections 1.1 or 1.1.1. So I suggest that we review the numbering of the subsections here and below to get on the same view.

It has been corrected in the revised manuscript as reviewer comment.

  1. Line 70: Water and the majority of organic solvents can dissolve chitin, a hydrophobic substance. Check this sentence. Can chitin dissolve in water? Also, the sentence is difficult to understand.

It has been corrected and rephrased in the revised manuscript as reviewer comment.

  1. Check the formatting of the captions to the pictures. It seems to me to be different in the different pictures.

It has ben checked and corrected in the revised manuscript as reviewer comment.

  1. Line 135: what is meant by anti-biological activity? In my opinion, we need to somehow decipher this concept. Does it mean that it kills all living things in general?

Ant-biological activity replaced by antimicrobial activity in the revised manuscript.

  1. Check the formatting throughout the text - some references are highlighted and some are not. Also check against the sample citation style: "*.[*]." Is a period before the square brackets necessary?

It has been checked and revised throughout the txt in the revised  manuscripts as reviewer comment.

  1. Line 328: Surface nanostructures may produce extraordinary phenomena, such as the lotus effect (self-cleaning effect). Remove word repetition.

It the words repetition has been deleted in the revised manuscript as reviewer comment.

  1. "A high-voltage power source, a syringe, and a collector electrode may be used to construct a standard electrospinning apparatus that can be configured vertically or horizontally as illustrated in Fig. 6." Maybe Figure 7, not 6?

It has been corrected in the revised manuscript as reviewer comment.

  1. Why is section 8 in red font? You should have carefully reviewed and formatted the article before submitting it to the journal.

It was revied in red colour as comment from previous reviewer and now It has been corrected in the revised manuscript as reviewer comment.

  1. Among these properties are the functional groups of nanofibrous polymers, which can coordinate with other components to promote cell adhesion, proliferation, differentiation, and, eventually, tissue regeneration. (Sofi, Ashraf, Beigh, & Sheikh, 2018). Reference in a wrong format.

It has been corrected in the revised manuscript as reviewer comment.

  1. After section 8.4.1 comes section 7.2 Why so?

It has been corrected in the revised manuscript as reviewer comment.

  1. Conclusions section: 5. Conclusion. Why 5? 

It has been corrected in the revised manuscript as reviewer comment.

  1. Section 6. References. Redo the numbers.

It has been corrected in the revised manuscript as reviewer comment.

Round 2

Reviewer 1 Report (New Reviewer)

Well done, author, in revising the paper. Congratulations

Check the grammatical errors

Author Response

Comments and Suggestions for Authors: Well done, author, in revising the paper. Congratulations

Thank you for your response.

Comments on the Quality of English Language: Check the grammatical errors.

The revised manuscript has been checked and corrected for grammatical errors as reviewer comment.

Reviewer 2 Report (New Reviewer)

The manuscript has been sufficiently improved for publication in Polymers. 

The quality of English language has been sufficiently improved for publication in Polymers. 

Author Response

Comments and Suggestions for Authors: The manuscript has been sufficiently improved for publication in Polymers. 

Thank you for your response.

Comments on the Quality of English Language: The quality of English language has been sufficiently improved for publication in Polymers..

Thank you for your response.

Reviewer 3 Report (New Reviewer)

The authors must use the same terminology entire whole article (ex: fibers or fibres??).

-

Author Response

Comments and Suggestions for Authors

The authors must use the same terminology entire whole article (ex: fibers or fibres??)

The revised manuscript has been checked and corrected as reviewer comment.

Reviewer 4 Report (New Reviewer)

Thank you for the corrections. But I also found more:

1. Check the captions to the pictures. You have two figures in your article with the caption "figure 2"

2. Figure 3 is divided into subdrawings a-d. You need to correct the caption. There is no transcript for a-c. Figure 3d needs to be attached to figures 3 a-c, or named with the number 4. Figure 3b,c needs to be redone. Its appearance does not match the level of this journal. 

3. Why first comes the title "Table 3. summarize the biological application of chitosan and its derivatives" and then the title "Table 1"?  Also check the formatting of the title of Table 3.

4. Again the appearance of "Table 3. Tissue engineering applications of chitosan nanofibers". Correct the numbering. 

5. Check the numbering 8. Conclusion, 10. References. Where is the number 9?

Author Response

Reviewer 4

Comments and Suggestions for Authors

Thank you for the corrections. But I also found more:

  1. Check the captions to the pictures. You have two figures in your article with the caption "figure 2".

It has been checked and corrected in the revised manuscript as reviewer comments.

  1. Figure 3 is divided into subdrawings a-d. You need to correct the caption. There is no transcript for a-c. Figure 3d needs to be attached to figures 3 a-c, or named with the number 4. Figure 3b,c needs to be redone. Its appearance does not match the level of this journal.

It has been checked and corrected in the revised manuscript as reviewer comments.

  1. Why first comes the title "Table 3. summarize the biological application of chitosan and its derivatives" and then the title "Table 1"?  Also check the formatting of the title of Table 3.

It has been checked and corrected in the revised manuscript as reviewer comments.

  1. Again the appearance of "Table 3. Tissue engineering applications of chitosan nanofibers". Correct the numbering.

It has been checked and corrected in the revised manuscript as reviewer comments.

  1. Check the numbering 8. Conclusion, 10. References. Where is the number 9

It has been checked and corrected in the revised manuscript as reviewer comments.

This manuscript is a resubmission of an earlier submission. The following is a list of the peer review reports and author responses from that submission.

Round 1

Reviewer 1 Report

Reviewer 1

Manuscript: polymers-2002486

Title: A review of chitosan and its nanofiber formation: preparation, characterization and its applications in wound dressing

 This revision deals with the preparation of chitosan nanofibers, followed by a discussion of the biocompatibility and degradation of chitosan nanofibers. Then it explains how to load a drug into the nanofibers. The authors also exemplify the uses of chitosan nanofibers in drug delivery systems. The subject of this review is interesting and in general has a good proposal for development of the subject. However, it is not well developed as the original proposal. Part 1 of the paper is really directed to chitin and chitosan, but part 2 (1.2 section) is too much generic for basically nanoparticles and nanomaterials (all types).  The third goal related to the uses of chitosan nanofibers in drug delivery systems were not well exemplified. It should be improved. In this sense, the paper should be carefully revised tanking into consideration the main goals of the subject proposed by the authors. I recommend a major revisor before it can be further considered for publication.

In addition, the manuscript must be improved in the English language aspects to become clearer for the reader. The text has many short paragraphs that are incomplete, and many problems related to grammar and usage. I believe that after a thorough and complete revision of the manuscript, the text will be much more interesting for the reader. Below, I give some of the points that I observed, but they are not complete. These are only a few examples.

Introduction

Line 40: “Chitin hydrolyzed in alkaline medium in 1894 to produce new carbohydrate soluble in acid called chitosan.” Please, improve the statement.

Line 44: “poly N-acetamido-2-decoxy-B-D-glucoce”; Should correct to poly N-acetamido-2-decoxy-Beta (b)-D-glucoce

Line 45: “Chitosan deacetylated chitin prepared by alkaline hydrolysis of chitin (Figure. 1)” Is this out of context? Something is missing connecting the earlier phrase.

Chitosan is a natural cationic polymer produced by deacetylation of chitin, is it right?

Line 66: “1.1.2.2. Molecular Weight (M. WT)”. The actual term recommended by IUPAC is Molar Mass.

Line 81: “As we see chitin and chitosan has many vestries properties but their usage are limited due to it is poor solvent, has low surface area and porous material.” Please, rewrite this sentence.

Line 210: “Applications of chitosan and chitosan derivatives” (line 210 – 238)

The general applications are listed, but more critical details should be provided.

Line 239: “1.2. Nanoparticles and Nontechnology” (line 239 – 545).

This part of the paper is interesting, but I believe is out of the proposed scope of the paper (either in the Abstract or in the Introduction). This should be used to cover the preparation and application of nanomaterials based on chitin or chitosan.

 Conclusion

Line 762: “Preparation, fibre configuration, material selection, intended applications, and spinning method are all aspects of electrospinning.” Please, rewrite this sentence.

Line 763: “Chitosan-based platform chemicals are expected to be increasingly used in biological applications in the future. However, our understanding of chitosan-based nanoparticles is restricted. Its biological properties and production should be studied further.” What do you mean about synthesis of chitosan?

Figures: All figures not provided by the own authors should indicate the respective reference (e.g., Figure 2, 5, 6).

Figure 2: the resolution of this figure is not good.

Figure 3: the resolution of this figure is not good.

Figure 5: the resolution of this figure is not good.

Author Response

Manuscript title: A review of chitosan and its nanofiber formation: preparation,

characterization and its biological applications

Manuscript ID:   

 polymers-2002486-R1

            Authors would like to express their appreciation for the extremely thoughtful Editor and reviews. We have been able to revise and improve the manuscript as a result of reviewer’s valuable feedback.

Reviewer 1:

Manuscript: polymers-2002486

Title: A review of chitosan and its nanofiber formation: preparation, characterization and its applications in wound dressing

This revision deals with the preparation of chitosan nanofibers, followed by a discussion of the biocompatibility and degradation of chitosan nanofibers. Then it explains how to load a drug into the nanofibers. The authors also exemplify the uses of chitosan nanofibers in drug delivery systems. The subject of this review is interesting and in general has a good proposal for development of the subject. However, it is not well developed as the original proposal. Part 1 of the paper is really directed to chitin and chitosan, but part 2 (1.2 section) is too much generic for basically nanoparticles and nanomaterials (all types).  The third goal related to the uses of chitosan nanofibers in drug delivery systems were not well exemplified. It should be improved. In this sense, the paper should be carefully revised tanking into consideration the main goals of the subject proposed by the authors. I recommend a major revisor before it can be further considered for publication.

In addition, the manuscript must be improved in the English language aspects to become clearer for the reader. The text has many short paragraphs that are incomplete, and many problems related to grammar and usage. I believe that after a thorough and complete revision of the manuscript, the text will be much more interesting for the reader. Below, I give some of the points that I observed, but they are not complete. These are only a few examples.

Introduction

  1. Line 40: “Chitin hydrolyzed in alkaline medium in 1894 to produce new carbohydrate soluble in acid called chitosan.” Please, improve the statement.

It has been improved as reviewer comment in the revised manuscript.

  1. Line 44: “poly N-acetamido-2-decoxy-B-D-glucoce”; Should correct to poly N-acetamido-2-decoxy-Beta (b)-D-glucoce

It has been improved as reviewer comment in the revised manuscript.

  1. Line 45: “Chitosan deacetylated chitin prepared by alkaline hydrolysis of chitin (Figure. 1)” Is this out of context? Something is missing connecting the earlier phrase.

It has been improved as reviewer comment in the revised manuscript.

  1. Chitosan is a natural cationic polymer produced by deacetylation of chitin, is it right?

Chitosan contains amino groups and soluble in dilute acids. Therefore, it has cationic characters.

  1. Line 66: “1.1.2.2. Molecular Weight (M. WT)”. The actual term recommended by IUPAC is Molar Mass.

Although the actual term recommended by IUPAC is Molar mass but molecular weight term recommended for macromolecules to it is pure number can be used.

  1. Line 81: “As we see chitin and chitosan has many vestries properties but their usage are limited due to it is poor solvent, has low surface area and porous material.” Please, rewrite this sentence.

It has been rewritten as reviewer comment in the revised manuscript.

  1. Line 210: “Applications of chitosan and chitosan derivatives” (line 210 – 238)

The general applications are listed, but more critical details should be provided.

More specific detailed have been provided in the revised version.

  1. Line 239: “1.2. Nanoparticles and Nontechnology” (line 239 – 545).

This part of the paper is interesting, but I believe is out of the proposed scope of the paper (either in the Abstract or in the Introduction). This should be used to cover the preparation and application of nanomaterials based on chitin or chitosan.

It has been revised as reviewer comment

  1. Conclusion

Line 762: “Preparation, fibre configuration, material selection, intended applications, and spinning method are all aspects of electrospinning.” Please, rewrite this sentence.

It has been rewritten as reviewer comment in the revised manuscript.

Line 763: “Chitosan-based platform chemicals are expected to be increasingly used in biological applications in the future. However, our understanding of chitosan-based nanoparticles is restricted. Its biological properties and production should be studied further.” What do you mean about synthesis of chitosan?

It has been revised as reviewer comment in the revised manuscript.

Figures: All figures not provided by the own authors should indicate the respective reference (e.g., Figure 2, 5, 6).

All figure has been cited in the revised submission

Author Response

Manuscript title: A review of chitosan and its nanofiber formation: preparation,

characterization and its biological applications

Manuscript ID:   

 polymers-2002486-R1

            Authors would like to express their appreciation for the extremely thoughtful Editor and reviews. We have been able to revise and improve the manuscript as a result of reviewer’s valuable feedback.

Reviewer 2:

After careful reading of the review entitled “A review of chitosan and its nanofiber formation: preparation, characterization and its applications in wound dressing" by Ibrahim M.A. et.al., I recommend to reject it as a publication in Polymers journal. The article is written in a very chaotic manner, the information is repeated or very general. Some parts are completely unnecessary, such as the history of nanotechnology, methods of obtaining nanoparticles or data in Table 2. Most of the drawings appear to be replicate from other sources, without clearly referencing them.

Thank you very much for your comments. Th e review articles has been fully revised and unnecessary part such as history of nanotechnology, methods of obtaining nanoparticles or data in Table 2 have been removed from the revised submission. I addition all figures from other sources now cited with their reference.

Reviewer 3 Report

In this manuscript, the authors reviewed the A review of chitosan and its nanofiber formation: preparation, characterization, and its applications in wound dressing. In my opinion, some issues should be further addressed and I hope the following comments could be helpful in improving their paper.

  1. In the introduction, the background about wound dressing is little, the authors should enrich this part and emphasize the necessity of "chitosan and its nanofiber" for wound dressing
  2. The authors focused on wound dressing, but what are the distinguished properties and specific problems of wound dressing? The authors never discussed it.
  3. According to the applications, most, if not all are applicable to other kinds of diseases. Then why did the authors not expand the topic to another topic?
  4. Good quality figures are very important for a good review paper, try to improve the quality of fig in this manuscript. Try to add at least 4-5 figures in this manuscript.
  5. The authors should summarize the current approaches to wound dressing and compare their advantages and disadvantages in order to draw the reader's attention.
  6. This manuscript is well organized but lacks specific comparative analysis. What are the advantages of "chitosan and its nanofiber" compared with technology for wound dressing?
  7. Please revisit the entire manuscript for minor grammar issues.
  8. In conclusions and perspectives, the author should consider giving some methodological design about how to improve the performance of such "chitosan and its nanofiber formation".
  9. Kindly add references to tables

Author Response

Manuscript title: A review of chitosan and its nanofiber formation: preparation,

characterization and its biological applications

Manuscript ID:   

 polymers-2002486-R1

            Authors would like to express their appreciation for the extremely thoughtful Editor and reviews. We have been able to revise and improve the manuscript as a result of reviewer’s valuable feedback.

Reviewer 3.

In this manuscript, the authors reviewed the A review of chitosan and its nanofiber formation: preparation, characterization, and its applications in wound dressing. In my opinion, some issues should be further addressed and I hope the following comments could be helpful in improving their paper.

  1. In the introduction, the background about wound dressing is little, the authors should enrich this part and emphasize the necessity of "chitosan and its nanofiber" for wound dressing. The authors focused on wound dressing, but what are the distinguished properties and specific problems of wound dressing? The authors never discussed it.

More recent research studies added in the revised manuscript

  1. According to the applications, most, if not all are applicable to other kinds of diseases. Then why did the authors not expand the topic to another topic?

More recent research studies added in the revised manuscript

  1. Good quality figures are very important for a good review paper, try to improve the quality of fig in this manuscript. Try to add at least 4-5 figures in this manuscript.

More recent research studies added in the revised manuscript

  1. The authors should summarize the current approaches to wound dressing and compare their advantages and disadvantages in order to draw the reader's attention. This manuscript is well organized but lacks specific comparative analysis. What are the advantages of "chitosan and its nanofiber" compared with technology for wound dressing?

It has been revised as reviewer comments

  1. Please revisit the entire manuscript for minor grammar issues.

The whole manuscript has been checked for typos and grammar issue

  1. Kindly add references to tables

It has been added in the revised manuscript

Reviewer 4 Report

Comments

In this paper, the authors reviewed the preparation, characterization, and application of chitosan and its nanofiber formation. The content is rich and substantial. However, there are still some issues to be addressed. The specific comments can be found as following:

1.     There are some punctuation problems in the text. For example, there is no period at the end of the fourth paragraph in 1.1.1., there is no period at the end of the paragraph of 1.1.2.2., and there is a comma at the end of the paragraph of 1.1.2.3., etc.

2.     More background on chitosan should be provided with some recent and important articles: Recent advancements in applications of chitosan-based biomaterials for skin tissue engineering; New Ulva lactuca Algae Based Chitosan Bio-composites for Bioremediation of Cd(II) Ions; etc.

3.     Title 1.1.2. is missing, and the title format needs to be unified.

4.     There are two titles 1.5.2.

5.     The definition of Figure (2) and Figure (5) is too low, so the definition of the picture should be increased.

6.     Fig. 6. should be changed to Figure (8).

7.     Three line tables should be applied.

8.     There are too many paragraphs with one or two sentences, which should be combined into together with a more logic way.

9.     Many figures should be replaced or modified with better resolution and readability.

10. The logic of the whole manuscript should be rearranged, especially the outline.

11. Authors have introduced the chitosan nanofibers by electrospinning. More background on the electrospinning and its applications should be provided with supporting articles: Nanomaterials 10 (1), 150, 2020; Electrospun fibrous materials and their applications for electromagnetic interference shielding: A review; e-Polymers 20 (1), 682-712, 2020; A review of smart electrospun fibers toward textiles; Materials & Design 214, 110406, 2022; etc.

12. The challenges and possible solutions should be proposed in one more section of Future perspectives.

13. There are still some typos and grammar issues in the manuscript. Authors should carefully recheck the whole manuscript.

Author Response

Reviewer 4:

In this paper, the authors reviewed the preparation, characterization, and application of chitosan and its nanofiber formation. The content is rich and substantial. However, there are still some issues to be addressed. The specific comments can be found as following:

  1. There are some punctuation problems in the text. For example, there is no period at the end of the fourth paragraph in 1.1.1., there is no period at the end of the paragraph of 1.1.2.2., and there is a comma at the end of the paragraph of 1.1.2.3., etc.

It has been added in the revised submission.

  1. More background on chitosan should be provided with some recent and important articles: Recent advancements in applications of chitosan-based biomaterials for skin tissue engineering; New Ulva lactuca Algae Based Chitosan Bio-composites for Bioremediation of Cd(II) Ions; etc.

It has been added in the revised submission.

  1. Title 1.1.2. is missing, and the title format needs to be unified.

It has been added in the revised submission.

  1. There are two titles 1.5.2.

It has been added in the revised submission.

  1. Fig. 6. should be changed to Figure (8).

It has been revised in the revised manuscript

  1. Three-line tables should be applied.

Three-line tables have been applied in the revised submission.

  1. There are too many paragraphs with one or two sentences, which should be combined into together with a more logic way.

It has been revised and applied in the revised submission.

  1. Many figures should be replaced or modified with better resolution and readability.

It has been added in the revised submission.

  1. The logic of the whole manuscript should be rearranged, especially the outline.

Revised as reviewer comment

  1. Authors have introduced the chitosan nanofibers by electrospinning. More background on the electrospinning and its applications should be provided with supporting articles: Nanomaterials 10 (1), 150, 2020; Electrospun fibrous materials and their applications for electromagnetic interference shielding: A review; e-Polymers 20 (1), 682-712, 2020; A review of smart electrospun fibers toward textiles; Materials & Design 214, 110406, 2022; etc.

It has been revised and the suggested references have been added in the revised manuscript.

  1. There are still some typos and grammar issues in the manuscript. Authors should carefully recheck the whole manuscript.

The whole manuscript has been checked for typos and grammar issue

Round 2

Reviewer 1 Report

Reviewer 1

Manuscript: polymers-2002486 (version 2)

Title: A review of chitosan and its nanofiber formation: preparation, characterization and its applications in wound dressing

The revised version of the article is much better. Nonetheless, it still has many problems related to the English language, in my opinion. This is not a matter of be meticulous, but without this improvement everyone will lose something. The Journal, the reader, and the authors since future citations will be precluded by the absence of readers. Thus, the authors should seek a professional service for reding the work (which has good ideas and content) and correct the format and general usage of the language.

Author Response

The revised version of the article is much better. Nonetheless, it still has many problems related to the English language, in my opinion. This is not a matter of be meticulous, but without this improvement everyone will lose something. The Journal, the reader, and the authors since future citations will be precluded by the absence of readers. Thus, the authors should seek a professional service for reding the work (which has good ideas and content) and correct the format and general usage of the language.

An English-speaking colleague has polished the language in this manuscript.

Reviewer 2 Report

I stand by my opinion, the article is written in a chaotic manner, the information is repeated or very general. Some parts are completely unnecessary, such as the history of nanotechnology or described applications of inorganic nanoparticles (2.3) – no relevance to chitosan. The drawings are still poor quality, without  permission of reprinting or wrong citation e.g. reference for figure 2 is Calvo et al [28], whereas in line 119-120 the [28] reference is Alonso 119 et al. and in the reference list in other paper: Sathiyabama, M. and R. Parthasarathy, Biological preparation of chitosan nanoparticles and its in vitro antifungal efficacy against 898 some phytopathogenic fungi. Carbohydrate Polymers, 2016. 151: p. 321-325. Therefore bearing in mind the high standards of the Polymers journal I recommend to reject is as a publication.

Author Response

Reviewer 2

I stand by my opinion, the article is written in a chaotic manner, the information is repeated or very general. Some parts are completely unnecessary, such as the history of nanotechnology or described applications of inorganic nanoparticles (2.3) – no relevance to chitosan. The drawings are still poor quality, without permission of reprinting or wrong citation e.g., reference for figure 2 is Calvo et al [28], whereas in line 119-120 the [28] reference is Alonso 119 et al. and in the reference list in other paper: Sathiyabama, M. and R. Parthasarathy, Biological preparation of chitosan nanoparticles and it’s in vitro antifungal efficacy against some phytopathogenic fungi. Carbohydrate Polymers, 2016. 151: p. 321-325.

The manuscript has been carefully reviewed and revised as reviewer comments and all unnecessary parts described by the reviewer deleted in the revised manuscript. In addition, figure 2 now has permission from John Wiley and Sons. All other parts reviewed carefully.

Reviewer 3 Report

Accepted in present form 

Author Response

Reviewer 3

Accepted in present form.

Thank you for your response

Reviewer 4 Report

Authors have carefully respond the previous comments except one minor issues for the previous comment 2. Regarding that comment, authors should highlight the corresponding corrections in the revised manuscript, but not only say "The problem is solved". Please provide detailed responses. Therefore, a minor revision is suggested.

Author Response

Reviewer 4

Authors have carefully responded the previous comments except one minor issues for the previous comment 2. Regarding that comment, authors should highlight the corresponding corrections in the revised manuscript, but not only say "The problem is solved". Please provide detailed responses. Therefore, a minor revision is suggested.”” 2. More background on chitosan should be provided with some recent and important articles: Recent advancements in applications of chitosan-based biomaterials for skin tissue engineering;”

I sorry for that now these two references added to the revised manuscript with numbers 6 and 88. In addition more cited reference added to update the background of chitosan.

Round 3

Reviewer 1 Report

Reviewer 1

Manuscript: polymers-2002486 (version 3)

Title: A review of chitosan and its nanofiber formation: preparation, characterization and its applications in wound dressing

The revised version (third) of the article has improved in the English language. The main problem that I still observe is that the title of the article and its contend is not well developed. There are many parts that are unnecessary (e.g., history of nanotechnology) that are not really a goal of the article (taking into consideration the proposed title). The argument including “applications in would dressing” does not seem to be fully developed. Only a few examples are provided without a deeper analysis. The authors show more applications in general, and “would dressing” is only an example. There is an inconsistency in this sense. This should be corrected, changing the title of the article or developing in a deeper manner the suggested application (i.e., would dressing). The second choice is the right one, but this would take more time to be done. Nevertheless, it would be in accordance with the objective of the special issue, in my opinion.

Author Response

Thank you for advance. Now the title of the article has been changed to: A review of chitosan and its nanofiber formation: preparation, characterization and its potential applications to be fit with the content. All unnecessary part such as history of nanotechnology have been removed from the revised second version. Herein we choose the first choice to be fit with the special issue.

Round 4

Reviewer 1 Report

The change in the title accomodates the content of the revision. Thus, the paper can be accepted in this form.